# Improved Algorithms for Collaborative PAC Learning

**Huy Lê Nguyễn**
College of Computer and Information Science
Northeastern University
Boston, MA 02115
hu.nguyen@northeastern.edu

**Lydia Zakynthinou**
College of Computer and Information Science
Northeastern University
Boston, MA 02115
zakynthinou.l@northeastern.edu

## Abstract

We study a recent model of collaborative PAC learning where $k$ players with $k$ different tasks collaborate to learn a single classifier that works for all tasks. Previous work showed that when there is a classifier that has very small error on all tasks, there is a collaborative algorithm that finds a single classifier for all tasks and has $O((\ln(k))^2)$ times the worst-case sample complexity for learning a single task. In this work, we design new algorithms for both the realizable and the non-realizable setting, having sample complexity only $O(\ln(k))$ times the worst-case sample complexity for learning a single task. The sample complexity upper bounds of our algorithms match previous lower bounds and in some range of parameters are even better than previous algorithms that are allowed to output different classifiers for different tasks.

## 1 Introduction

There has been a lot of work in machine learning concerning learning multiple tasks simultaneously, ranging from multi-task learning [3, 4], to domain adaptation [10, 11], to distributed learning [2, 7, 14]. Another area in similar spirit to this work is meta-learning, where one leverages samples from many different tasks to train a single algorithm that adapts well to all tasks (see e.g. [8]).

In this work, we focus on a model of collaborative PAC learning, proposed by [5]. In the classic PAC learning setting introduced by [13], where PAC stands for *probably approximately correct*, the goal is to learn a task by drawing from a distribution of samples. The optimal classifier that achieves the lowest error on the task with respect to the given distribution is assumed to come from a concept class $\mathcal{F}$ of VC dimension $d$. The VC theorem [1] states that for any instance $m_{\epsilon,\delta} = O\left(\frac{1}{\epsilon}\left(d\ln\left(\frac{1}{\epsilon}\right) + \ln\left(\frac{1}{\delta}\right)\right)\right)$ labeled samples suffice to learn a classifier that achieves low error with probability at least $1 - \delta$, where the error depends on $\epsilon$.

In the collaborative model, there are $k$ players attempting to learn their own tasks, each task involving a different distribution of samples. The goal is to learn a single classifier that also performs well on all the tasks. One example from [5], which motivates this problem, is having $k$ hospitals with different patient demographics which want to predict the overall occurrence of a disease. In this case, it would be more fitting as well as cost efficient to develop and distribute a single classifier to all the hospitals. In addition, the requirement for a single classifier is imperative in settings where there are fairness concerns. For example, consider the case that the goal is to find a classifier that predicts loan defaults for a bank by gathering information from bank stores located in neighborhoods with diverse socioeconomic characteristics. In this setting, the samples provided by each bank store come from different distributions while it is desired to guarantee low error rates for all the neighborhoods. Again, in this setting, the bank should employ a single classifier among all the neighborhoods.

If each player were to learn a classifier for their task without collaboration, they would each have to draw a sufficient number of samples from their distribution to train their classifier. Therefore, solving $k$ tasks independently would require $k \cdot m_{\epsilon,\delta}$ samples in the worst case. Thus, we are interested in algorithms that utilize samples from all players and solve all $k$ tasks with sample complexity $o\left(\frac{k}{\epsilon}\left(d\ln\left(\frac{1}{\epsilon}\right) + \ln\left(\frac{1}{\delta}\right)\right)\right)$.

Blum et al. [5] give an algorithm with sample complexity $O\left(\frac{\ln^2(k)}{\epsilon}\left((d+k)\ln\left(\frac{1}{\epsilon}\right) + k\ln\left(\frac{1}{\delta}\right)\right)\right)$ for the realizable setting, that is, assuming the existence of a single classifier with zero error on all the tasks. They also extend this result by proving that a slightly modified algorithm returns a classifier with error $\epsilon$, under the relaxed assumption that there exists a classifier with error $\epsilon/100$ on all the tasks. In addition, they prove a lower bound showing that there is a concept class with $d = \Theta(k)$ where $\Omega\left(\frac{k}{\epsilon}\ln\left(\frac{k}{\delta}\right)\right)$ samples are necessary.

In this work, we give two new algorithms based on multiplicative weight updates which have sample complexities $O\left(\frac{\ln(k)}{\epsilon}\left(d\ln\left(\frac{1}{\epsilon}\right) + k\ln\left(\frac{k}{\delta}\right)\right)\right)$ and $O\left(\frac{1}{\epsilon}\ln\left(\frac{k}{\delta}\right)\left(d\ln\left(\frac{1}{\epsilon}\right) + k + \ln\left(\frac{1}{\delta}\right)\right)\right)$ for the realizable setting. Our first algorithm matches the sample complexity of [5] for the variant of the problem in which the algorithm is allowed to return different classifiers to the players and our second algorithm has the sample complexity almost matching the lower bound of [5] when $d = \Theta(k)$ and for typical values of $\delta$. Both are presented in Section 3. Independently of our work, [6] use the multiplicative weight update approach and achieve the same bounds as we do in that section.

Moreover, in Section 4, we extend our results to the non-realizable setting, presenting two algorithms that generalize the algorithms for the realizable setting. These algorithms learn a classifier with error at most $(2+\alpha)\mathtt{OPT} + \epsilon$ on all the tasks, where $\alpha$ is set to a constant value, and have sample complexities $O\left(\frac{\ln(k)}{\alpha^4\epsilon}\left(d\ln\left(\frac{1}{\epsilon}\right) + k\ln\left(\frac{k}{\delta}\right)\right)\right)$ and $O\left(\frac{1}{\alpha^4\epsilon}\ln\left(\frac{k}{\delta}\right)\left(d\ln\left(\frac{1}{\epsilon}\right) + k\ln\left(\frac{1}{\alpha}\right) + \ln\left(\frac{1}{\delta}\right)\right)\right)$. With constant $\alpha$, these sample complexities are the same as in the realizable case. Finally, we give two algorithms with *randomized* classifiers whose error probability over the random choice of the example *and the classifier's randomness* is at most $(1+\alpha)\mathtt{OPT}+\epsilon$ for all tasks. The sample complexities of these algorithms are $O\left(\frac{\ln(k)}{\alpha^3\epsilon^2}\left(d\ln\left(\frac{1}{\epsilon}\right) + k\ln\left(\frac{k}{\delta}\right)\right)\right)$ and $O\left(\frac{1}{\alpha^3\epsilon^2}\ln\left(\frac{k}{\delta}\right)\left((d+k)\ln\left(\frac{1}{\epsilon}\right) + \ln\left(\frac{1}{\delta}\right)\right)\right)$.

## 2  Model

In the traditional PAC learning model, there is a space of instances $\mathcal{X}$ and a set $\mathcal{Y} = \{0,1\}$ of possible labels for the elements of $\mathcal{X}$. A classifier $f : \mathcal{X} \to \mathcal{Y}$, which matches each element of $\mathcal{X}$ to a label, is called a *hypothesis*. The error of a hypothesis with respect to a distribution $D$ on $\mathcal{X} \times \mathcal{Y}$ is defined as $\mathrm{err}_D(f) = \mathrm{Pr}_{(x,y)\sim D}[f(x) \neq y]$. Let $\mathtt{OPT} = \inf_{f \in \mathcal{F}}\mathrm{err}_D(f)$, where $\mathcal{F}$ is a class of hypotheses. In the realizable setting we assume that there exists a target classifier with zero error, that is, there exists $f^* \in \mathcal{F}$ with $\mathrm{err}_D(f^*) = \mathtt{OPT} = 0$ for all $i \in [k]$. Given parameters $(\epsilon, \delta)$, the goal is to learn a classifier that has error at most $\epsilon$, with probability at least $1 - \delta$. In the non-realizable setting, the optimal classifier $f^*$ is defined to have $\mathrm{err}_D(f^*) \leq \mathtt{OPT} + \varepsilon$ for any $\varepsilon > 0$. Given parameters $(\epsilon, \delta)$ and a new parameter $\alpha$, which can be considered to be a constant, the goal is to learn a classifier that has error at most $(1 + \alpha)\mathtt{OPT} + \epsilon$, with probability at least $1 - \delta$.

By the VC theorem and its known extension, the desired guarantee can be achieved in both settings by drawing a set of samples of size $m_{\epsilon,\delta} = O\left(\frac{1}{\epsilon}\left(d\ln\left(\frac{1}{\epsilon}\right) + \ln\left(\frac{1}{\delta}\right)\right)\right)$ and returning the classifier with minimum error on that sample. More precisely, in the non-realizable setting, $m_{\epsilon,\delta} = \frac{C}{\epsilon\alpha}\left(d\ln\left(\frac{1}{\epsilon}\right) + \ln\left(\frac{1}{\delta}\right)\right)$, where $C$ is also a constant. We consider an algorithm $\mathcal{O}_{\mathcal{F}}(S)$, where $S$ is a set of samples drawn from an arbitrary distribution $D$ over the domain $\mathcal{X} \times \{0,1\}$, that returns a hypothesis $f_0$ whose error on the sample set satisfies $\mathrm{err}_S(f_0) \leq \inf_{f \in \mathcal{F}}\mathrm{err}_S(f) + \varepsilon$ for any $\varepsilon > 0$, if such a hypothesis exists. The VC theorem guarantees that if $|S| = m_{\epsilon,\delta}$, then $\mathrm{err}_D(f_0) \leq (1+\alpha)\mathrm{err}_S(f_0) + \epsilon$.

In the collaborative model, there are $k$ players with distributions $D_1, \ldots, D_k$. Similarly, $\mathtt{OPT} = \inf_{f \in \mathcal{F}}\max_{i \in [k]}\mathrm{err}_{D_i}(f)$ and the goal is to learn a single good classifier for all distributions. In [5], the

authors consider two variants of the model for the realizable setting, the *personalized* and the *centralized*. In the former the algorithm can return a different classifier to each player, while in the latter it must return a single good classifier. For the personalized variant, Blum et al. give an algorithm with almost the same sample complexity as the lower bound they provide. We focus on the more restrictive centralized variant of the model, for which the algorithm that Blum et al. give does not match the lower bound. We note that the algorithms we present are *improper*, meaning that the classifier they return is not necessarily in the concept class $\mathcal{F}$.

## 3 Sample complexity upper bounds for the realizable setting

In this section, we present two algorithms and prove their sample complexity.

Both algorithms employ multiplicative weight updates, meaning that in each round they find a classifier with low error on the weighted mixture of the distributions and double the weights of the players for whom the classifier did not perform well. In this way, the next sample set drawn will include more samples from these players' distributions so that the next classifier will perform better on them. To identify the players for whom the classifier of the round did not perform well, the algorithms test the classifier on a small number of samples drawn from each player's distribution. If the error of the classifier on the sample is low, then the error on the player's distribution can not be too high and vise versa. In the end, both algorithms return the majority function over all the classifiers of the rounds, that is, for each point $x \in \mathcal{X}$, the label assigned to $x$ is the label that the majority of the classifiers assign to $x$.

We note that for typical values of $\delta$, Algorithm R2 is better than Algorithm R1. However, Algorithm R1 is always better than the algorithm of [5] for the centralized variant of the problem and matches their number of samples in the personalized variant, so we present both algorithms in this section. In the algorithms of [5], the players are divided into classes based on the number of rounds for which that player's task is not solved with low error. The number of classes could be as large as the number of rounds, which is $\Theta(\log(k))$, and their algorithm uses roughly $m_{\epsilon,\delta}$ samples from each class. On the other hand, Algorithm R1 uses only $m_{\epsilon,\delta}$ samples across all classes and saves a factor of $\Theta(\log(k))$ in the sample complexity. This requires analyzing the change in all classes together as opposed to class by class.

---

**Algorithm** R1
---

    **Initialize:** $\forall i \in [k]\ w_i^{(0)} := 1;\ t := 5\lceil \log(k) \rceil;\ \epsilon' := \epsilon/6;\ \delta' := \delta/(3t);$
    **for** $r = 1$ **to** $t$ **do**
        $\tilde{D}^{(r-1)} \leftarrow \frac{1}{\Phi^{(r-1)}} \sum_{i=1}^{k} \left( w_i^{(r-1)} D_i \right)$, where $\Phi^{(r-1)} = \sum_{i=1}^{k} w_i^{(r-1)};$
        Draw a sample set $S^{(r)}$ of size $m_{\epsilon'/16,\delta'}$ from $\tilde{D}^{(r-1)};$
        $f^{(r)} \leftarrow \mathcal{O}_{\mathcal{F}}(S^{(r)});$
        $G_r \leftarrow \text{TEST}(f^{(r)}, k, \epsilon', \delta');$
        **Update:** $w_i^{(r)} = \begin{cases} 2w_i^{(r-1)}, & \text{if } i \notin G_r \\ w_i^{(r-1)}, & \text{otherwise} \end{cases};$
    **end for**
    **return** $f_{\text{R1}} = \text{maj}(\{f^{(r)}\}_{r=1}^{t})$

    **Procedure** $\text{TEST}(f^{(r)}, k, \epsilon', \delta')$
    **for** $i = 1$ **to** $k$ **do**
        Draw a sample set $T_i$ of size $O\left( \frac{1}{\epsilon'} \ln\left( \frac{k}{\delta'} \right) \right)$ from $D_i;$
    **end for**
    **return** $\{i \mid \text{err}_{T_i}(f^{(r)}) \leq \frac{3}{4}\epsilon'\};$

---

Algorithm R1 runs for $t = \Theta(\log(k))$ rounds and learns a classifier $f^{(r)}$ in each round $r$ that has low error on the weighted mixture of the distributions $\tilde{D}^{(r-1)}$. For each player at least $0.6t$ of the learned classifiers are "good", meaning that they have error at most $\epsilon' = \epsilon/6$ on the player's distribution. Since the algorithm returns the majority of the classifiers, in order for an instance to be mislabeled, at

least $0.5t$ of the total number of classifiers should mislabel it. This implies that at least $0.1t$ of the "good" classifiers of that player should mislabel it, which amounts to $1/6$ of the "good" classifiers. Therefore, the error of the majority of the functions for that player is at most $6\epsilon' = \epsilon$.

To identify the players for whom the classifier of the round does not perform well, Algorithm R1 uses a procedure called TEST. This procedure draws $O\left(\frac{1}{\epsilon'} \ln\left(\frac{k}{\delta'}\right)\right)$ samples from each player's distribution and tests the classifier on these samples. If the error for a player's sample set is at most $3\epsilon'/4$ then TEST concludes that the classifier is good for that player and adds them to the returned set $G_r$. The samples that the TEST requires from each player suffice to make it capable of distinguishing between the players with error more than $\epsilon'$ and players with error at most $\epsilon'/2$ with respect to their distributions, with high probability.

**Theorem 1.** *For any $\epsilon, \delta \in (0, 1)$, and hypothesis class $\mathcal{F}$ of VC dimension d, Algorithm R1 returns a classifier $f_{R1}$ with $err_{D_i}(f_{R1}) \leq \epsilon \; \forall i \in [k]$ with probability at least $1 - \delta$ using $m$ samples, where*

$$m = O\Big(\frac{\ln(k)}{\epsilon}\Big(d \ln\Big(\frac{1}{\epsilon}\Big) + k \ln\Big(\frac{k}{\delta}\Big)\Big)\Big).$$

The proof of Theorem 1 is very similar to the one for Algorithm R2 so we omit it and refer the reader to Appendix A. Algorithm R1 is the natural boosting alternative to the algorithm of [5] for the centralized variant of the model. Although it is discussed in [5] and mentioned to have the same sample complexity as their algorithm, it turns out that it is more efficient. Its sample complexity is slightly better (or the same, depending on the parameter regime) compared to the one of the algorithm for the personalized setting presented in [5], which is $O\left(\frac{\log(k)}{\epsilon}\Big((d+k) \ln\Big(\frac{1}{\epsilon}\Big) + k \ln\Big(\frac{k}{\delta}\Big)\Big)\right)$.

However, in the setting of the lower bound in [5] where $k = \Theta(d)$, there is a gap of $\log(k)$ multiplicatively between the sample complexity of Algorithm R1 and the lower bound. This difference stems from the fact that in every round, the algorithm uses roughly $\Theta(k)$ samples to find a classifier but roughly $\Theta(k \log(k))$ samples to test the classifier for $k$ tasks. Motivated by this discrepancy, we develop Algorithm R2, which is similar to Algorithm R1 but uses fewer samples to test the performance of each classifier on the players' distributions. To achieve high success probability, Algorithm R2 uses a higher number of rounds.

---

**Algorithm** R2

---

**Initialize:** $\forall i \in [k] \; w_i^{(0)} := 1; \; t := 150 \left\lceil \log\left(\frac{k}{\delta}\right) \right\rceil; \; \epsilon' := \epsilon/6; \; \delta' := \delta/(4t);$
**for** $r = 1$ **to** $t$ **do**
$\quad \tilde{D}^{(r-1)} \leftarrow \frac{1}{\Phi^{(r-1)}} \sum_{i=1}^{k} \left(w_i^{(r-1)} D_i\right)$, where $\Phi^{(r-1)} = \sum_{i=1}^{k} w_i^{(r-1)};$
$\quad$ Draw a sample set $S^{(r)}$ of size $m_{\epsilon'/16, \delta'}$ from $\tilde{D}^{(r-1)};$
$\quad f^{(r)} \leftarrow \mathcal{O}_{\mathcal{F}}(S^{(r)});$
$\quad G_r \leftarrow \text{FASTTEST}(f^{(r)}, k, \epsilon', \delta');$
$\quad$ **Update:** $w_i^{(r)} = \begin{cases} 2w_i^{(r-1)}, & \text{if } i \notin G_r \\ w_i^{(r-1)}, & \text{otherwise} \end{cases};$
**end for**
**return** $f_{R2} = \text{maj}(\{f^{(r)}\}_{r=1}^{t});$

**Procedure** FASTTEST$(f^{(r)}, k, \epsilon', \delta')$
**for** $i = 1$ **to** $k$ **do**
$\quad$ Draw a sample set $T_i$ of size $O\left(\frac{1}{\epsilon'}\right)$ from $D_i;$
**end for**
**return** $\{i \mid err_{T_i}(f^{(r)}) \leq \frac{3}{4}\epsilon'\};$

---

More specifically, Algorithm R2 runs for $t = 150\lceil \log(\frac{k}{\delta})\rceil$ rounds. In addition, the test it uses to identify the players for whom the classifier of the round does not perform well requires $O\left(\frac{1}{\epsilon'}\right)$ samples from each player. This helps us save one logarithmic factor in the second term of the sample complexity of Algorithm R1. We call this new test FASTTEST. The fact that FASTTEST uses less

samples causes it to be less successful at distinguishing the players for whom the classifier was "good" from the players for whom it was not, meaning that it has constant probability of making a mistake for a given player at a given round. There are two types of mistakes that FASTTEST can make: to return $i \notin G_r$ and double the weight of $i$ when the classifier is good for $i$'s distribution and to return $i \in G_r$ and not double the weight of $i$ when the classifier is not good.

**Theorem 2.** *For any $\epsilon, \delta \in (0, 1)$, and hypothesis class $\mathcal{F}$ of VC dimension $d$, Algorithm R2 returns a classifier $f_{R2}$ with $err_{D_i}(f_{R2}) \leq \epsilon \ \forall i \in [k]$ with probability at least $1 - \delta$ using $m$ samples, where*

$$m = O\left(\frac{1}{\epsilon} \ln\left(\frac{k}{\delta}\right)\left(d \ln\left(\frac{1}{\epsilon}\right) + k + \ln\left(\frac{1}{\delta}\right)\right)\right).$$

To prove the correctness and sample complexity of Algorithm R2, we need Lemma 2.1, which describes the set $G_r$ that the FASTTEST returns. Its proof uses the multiplicative forms of the Chernoff bounds and is in Appendix A.

**Lemma 2.1.** *FASTTEST($f^{(r)}, k, \epsilon', \delta'$) is such that the following two properties hold, each with probability at least $0.99$, for given round $r \in [t]$ and player $i \in [k]$.*

*(a) If $err_{D_i}(f^{(r)}) > \epsilon'$, then $i \notin G_r$.*

*(b) If $err_{D_i}(f^{(r)}) \leq \frac{\epsilon'}{2}$, then $i \in G_r$.*

*Proof of Theorem 2.* First, we prove that Algorithm R2 indeed learns a good classifier, meaning that, with probability at least $1 - \delta$, for every player $i \in [k]$ the returned classifier $f_{R2}$ has error $err_{D_i}(f_{R2}) \leq \epsilon$. Let $e_i^{(t)}$ be the number of rounds for which the classifier's error on $D_i$ was more than $\epsilon'$, i.e. $e_i^{(t)} = |\{r \mid r \in [t] \text{ and } err_{D_i}(f^{(r)}) > \epsilon'\}|$.

**Claim 2.1.** *With probability at least $1 - \delta$, $e_i^{(t)} < 0.4t \ \forall i \in [k]$.*

If the claim holds, then with probability at least $1 - \delta$, less than $0.4t$ functions have error more than $\epsilon'$ on $D_i$, $\forall i \in [k]$. Therefore, with probability at least $1 - \delta$, $err_{D_i}(f_{R2}) \leq \frac{0.6}{0.1}\epsilon' \leq \epsilon$ for every $i \in [k]$.

*Proof of Claim 2.1.* Let us denote by $I^{(r)}$ the set of players having $err_{D_i}(f^{(r)}) > \frac{\epsilon'}{2}$ in round $r$, i.e., $I^{(r)} = \{i \in [k] \mid err_{D_i}(f^{(r)}) > \frac{\epsilon'}{2}\}$. We condition on the randomness in the first $r - 1$ rounds and compute $\mathbb{E}[\Phi^{(r)} \mid \Phi^{(r-1)}]$. By linearity of expectation, the following hold for round $r$:

$$err_{\tilde{D}^{(r-1)}}(f^{(r)}) = \frac{1}{\Phi^{(r-1)}}\sum_{i=1}^{k}\left(w_i^{(r-1)}err_{D_i}(f^{(r)})\right) \geq \frac{1}{\Phi^{(r-1)}}\sum_{i \in I^{(r)}\setminus G_r}\left(w_i^{(r-1)}err_{D_i}(f^{(r)})\right) \tag{1}$$

By the definition of $I^{(r)}$, $err_{D_i}(f^{(r)}) > \frac{\epsilon'}{2}$ for $i \in I^{(r)}$. From the VC theorem, with probability at least $1 - \delta'$, $err_{\tilde{D}^{(r-1)}}(f^{(r)}) \leq \frac{\epsilon'}{16}$. Using these two bounds and inequality (1), it follows that with probability at least $1 - \delta'$,

$$\sum_{i \in I^{(r)}\setminus G_r} w_i^{(r-1)} \leq \frac{1}{8}\Phi^{(r-1)}. \tag{2}$$

For the rest of the analysis, we will condition our probability space to the event that inequality (2) holds for all $t$ rounds. By the union bound, this event happens with probability $1 - t\delta' = 1 - \delta/4$.

Consider the set of players $i \notin I^{(r)} \cup G_r$. These are the players for whom the classifier of the round performed well but FASTTEST made a mistake and did not include them in the set $G_r$. By linearity of expectation:

$$\mathbb{E}[\sum_{i \notin G_r} w_i^{(r-1)} \mid \Phi^{(r-1)}] = \mathbb{E}\left[\sum_{i \in I^{(r)}\setminus G_r} w_i^{(r-1)} + \sum_{i \notin I^{(r)}\cup G_r} w_i^{(r-1)} \Bigg| \Phi^{(r-1)}\right] \tag{3}$$
$$\underset{\text{(2), Lemma 2.1(b)}}{\leq} (0.125 + 0.01)\Phi^{(r-1)}$$

Thus, the expected value of the potential function in round $r$ conditioned on its value in the previous round is bounded by

$$\mathbb{E}[\Phi^{(r)} \mid \Phi^{(r-1)}] = \mathbb{E}\left[\sum_{i=1}^{k} w_i^{(r-1)} + \sum_{i \notin G_r} w_i^{(r-1)} \middle| \Phi^{(r-1)}\right] \overset{(3)}{\leq} 1.135\Phi^{(r-1)}.$$

By the definition of the expected value, this implies that $\mathbb{E}[\Phi^{(r)}] \leq 1.135\,\mathbb{E}[\Phi^{(r-1)}]$. Conditioned on the fact that inequality (2) holds for all rounds, which is true with probability at least $1 - \frac{\delta}{4}$, we can conclude that $\mathbb{E}[\Phi^{(t)}] \leq k(1.135)^t$, by induction. Using Markov's inequality we can state that $\Pr\left[\Phi^{(t)} \geq \frac{\mathbb{E}[\Phi^{(t)}]}{\delta/2}\right] \leq \delta/2$. It follows that with probability at least $1 - \frac{\delta}{4} - \frac{\delta}{2} = 1 - \frac{3\delta}{4}$

$$\Phi^{(t)} \leq \frac{2k(1.135)^t}{\delta}. \tag{4}$$

We now need a lower bound for $w_i^{(t)}$. Let $m_i^{(r)}$ denote the number of rounds $r'$, up until and including round $r$, for which the procedure FASTTEST made a mistake and returned $i \in G_{r'}$ although $\mathrm{err}_{D_i}(f^{(r')}) > \epsilon'$. From Lemma 2.1(a), it follows that $\mathbb{E}[m_i^{(r)} - m_i^{(r-1)}] \leq 0.01$ so for $M_i^{(r)} = m_i^{(r)} - 0.01r$ it holds that $\mathbb{E}[M_i^{(r)} \mid M_i^{(r-1)}] \leq M_i^{(r-1)}$. Therefore, the sequence $\{M_i^{(r)}\}_{r=0}^{t}$ is a super-martingale. In addition to this, since we can make at most one mistake in each round, it holds that $M_i^{(r)} - M_i^{(r-1)} < 1$. Using the Azuma-Hoeffding inequality with $M_i^{(0)} = m_i^{(0)} - 0.01 \cdot 0 = 0$ and the fact that $t \geq 150$ we calculate that

$$\Pr\left[m_i^{(t)} \geq 0.18t\right] \leq \exp\left(-\frac{(0.17t)^2}{2t}\right) \leq \frac{\delta}{4k}.$$

By union bound, $m_i^{(t)} < 0.18t$ holds $\forall i \in [k]$ with probability at least $1 - \frac{\delta}{4}$.

The number of times a weight is doubled throughout the algorithm is $\log(w_i^{(t)})$ and it is at least the number of times the error of the classifier was more than $\epsilon'$ minus the number of times the error was more than $\epsilon'$ but the FASTTEST made a mistake, which is exactly $e_i^{(t)} - m_i^{(t)}$. So $w_i^{(t)} \geq 2^{e_i^{(t)} - m_i^{(t)}} > 2^{e_i^{(t)} - 0.18t}$ holds for all $i \in [k]$ with probability at least $1 - \frac{\delta}{4}$. Combining this with the bound from inequality (4) we have that with probability at least $1 - \delta$:

$$w_i^{(t)} \leq \Phi^{(t)} \Rightarrow 2^{e_i^{(t)} - 0.18t} < \frac{2k(1.135)^t}{\delta} \Rightarrow e_i^{(t)} - 0.18t < 1 + \log\left(\frac{k}{\delta}\right) + t\log(1.135)$$

$$\Rightarrow e_i^{(t)} < 0.18t + \frac{1}{150}t + \frac{1}{150}t + 0.183t < 0.4t \qquad \blacksquare$$

It remains to bound the number of samples. FASTTEST is called $t = 150\lceil\log(\frac{k}{\delta})\rceil$ times, so it requires $O\left(\log\left(\frac{k}{\delta}\right)\frac{k}{\epsilon'}\right) = O\left(\frac{k}{\epsilon}\log\left(\frac{k}{\delta}\right)\right)$ samples in total. The number of samples required to learn each round's classifier is $m_{\epsilon'/16, \delta'}$, so for all rounds there are required $O\left(\log\left(\frac{k}{\delta}\right)\frac{1}{\epsilon'}\left(d\ln\left(\frac{1}{\epsilon'}\right) + \ln\left(\frac{1}{\delta'}\right)\right)\right)$ samples. Substituting $\epsilon' = \epsilon/6$ and $\delta' = \delta/(4t) = \delta/\left(600\lceil\log\left(\frac{k}{\delta}\right)\rceil\right)$ we get $O\left(\frac{1}{\epsilon}\log\left(\frac{k}{\delta}\right)\left(d\ln\left(\frac{1}{\epsilon}\right) + \ln\left(\frac{\log(k)}{\delta}\right)\right)\right)$ samples in total. From the addition of the two bounds above, the overall sample complexity bound is:

$$O\left(\frac{1}{\epsilon}\ln\left(\frac{k}{\delta}\right)\left(d\ln\left(\frac{1}{\epsilon}\right) + k + \ln\left(\frac{1}{\delta}\right)\right)\right)$$

$\square$

## 4 Sample complexity upper bounds for the non-realizable setting

We design Algorithms NR1 and NR2 for the non-realizable setting, which generalize the results of Algorithms R1 and R2, respectively.

**Theorem 3.** *For any $\epsilon, \delta \in (0,1)$, $7\epsilon/6 < \alpha < 1$, and hypothesis class $\mathcal{F}$ of VC dimension $d$, Algorithm NR1 returns a classifier $f_{NR1}$ such that $\mathrm{err}_{D_i}(f_{NR1}) \leq (2+\alpha)\mathtt{OPT} + \epsilon$ holds for all $i \in [k]$ with probability $1 - \delta$ using $m$ samples, where*

$$m = O\Big(\frac{\ln(k)}{\alpha^4 \epsilon}\Big(d\ln\Big(\frac{1}{\epsilon}\Big) + k\ln\Big(\frac{k}{\delta}\Big)\Big)\Big).$$

**Theorem 4.** *For any $\epsilon, \delta \in (0,1)$, $5\epsilon/4 < \alpha < 1$, and hypothesis class $\mathcal{F}$ of VC dimension $d$, Algorithm NR2 returns a classifier $f_{NR2}$ such that $\mathrm{err}_{D_i}(f_{NR2}) \leq (2+\alpha)\mathtt{OPT} + \epsilon$ holds for all $i \in [k]$ with probability $1 - \delta$ using $m$ samples, where*

$$m = O\Big(\frac{1}{\alpha^4 \epsilon}\ln\Big(\frac{k}{\delta}\Big)\Big(d\ln\Big(\frac{1}{\epsilon}\Big) + k\ln\Big(\frac{1}{\alpha}\Big) + \ln\Big(\frac{1}{\delta}\Big)\Big)\Big).$$

Their main modification compared to the algorithms in the previous section is that these algorithms use a smoother update rule. Algorithm NR2 is presented here and Algorithm NR1 is in Appx. B.1.

---

**Algorithm** NR2

---

1: **Initialization**: $\forall i \in [k]\ w_i^{(0)} := 1$; $\alpha' := \alpha/40$; $t := 2\lceil \ln(4k/\delta)/\alpha'^3 \rceil$; $\epsilon' := \epsilon/64$; $\delta' := \delta/(4t)$;
2: **for** $r = 1, \dots, t$ **do**
3:     $\tilde{D}^{(r-1)} \leftarrow \frac{1}{\Phi^{(r-1)}}\sum_{i=1}^{k}\Big(w_i^{(r-1)}D_i\Big)$, where $\Phi^{(r-1)} := \sum_{i=1}^{k} w_i^{(r-1)}$;
4:     Draw a sample set $S^{(r)}$ of size $O\Big(\frac{1}{\alpha'\epsilon'}\Big(d\ln\Big(\frac{1}{\epsilon'}\Big) + \ln\Big(\frac{1}{\delta'}\Big)\Big)\Big)$ from $\tilde{D}^{(r-1)}$;
5:     $f^{(r)} \leftarrow \mathcal{O}_\mathcal{F}(S^{(r)})$;
6:     **for** $i = 1, \dots, k$ **do**
7:         Draw a sample set $T_i$ of size $O\Big(\frac{1}{\alpha'\epsilon'}\ln\Big(\frac{1}{\alpha'}\Big)\Big)$ from $D_i$;
8:         $s_i^{(r)} \leftarrow \min\Big(\frac{\mathrm{err}_{T_i}(f^{(r)})\alpha'^2}{(1+3\alpha')\mathrm{err}_{S^{(r)}}(f^{(r)})+3\epsilon'}, \alpha'\Big)$
9:         **Update**: $w_i^{(r)} \leftarrow w_i^{(r-1)}(1 + s_i^{(r)})$
10:    **end for**
11: **end for**
12:
13: **return** $f_{\mathrm{NR2}} = \mathrm{maj}(\{f^{(r)}\}_{r=1}^t)$;

---

Algorithm NR2 faces a similar challenge as Algorithm R2. Given a player $i$, since the number of samples $T_i$ used to estimate $\mathrm{err}_{D_i}(f^{(r)})$ in each round is low, the estimation is not very accurate. Ideally, we would want the inequality

$$|\mathrm{err}_{T_i}(f^{(r)}) - \mathrm{err}_{D_i}(f^{(r)})| \leq \alpha' \cdot \mathrm{err}_{D_i}(f^{(r)}) + \epsilon'$$

to hold for all players and all rounds with high probability. The "good" classifiers are now defined as the ones corresponding to rounds for which the inequality holds and $\mathrm{err}_{T_i}(f^{(r)})$ is not very high (an indication of which is that $s_i^{(r)} < \alpha'$). The expected number of rounds that either one of these properties does not hold is a constant fraction of the rounds ($\approx t\alpha'$) and due to the high number of rounds it is concentrated around that value, as in Algorithm R2. The proof of Theorem 4 is in Appx. B.2.

We note that the classifiers returned by these algorithms have a multiplicative approximation factor of almost 2 on the error. A different approach would be to allow for randomized classifiers with low error probability over both the randomness of the example and the classifier. We design two algorithms, NR1-AVG and NR2-AVG that return a classifier which satisfies this form of guarantee on the error without the 2-approximation factor but use roughly $\frac{\alpha}{\epsilon}$ times more samples. The returned classifier is a randomized algorithm that, given an element $x$, chooses one of the classifiers of all rounds uniformly at random and returns the label that this classifier gives to $x$. For any distribution over examples, the error probability of this randomized classifier is exactly the average of the error probability of classifiers $f^{(1)}, f^{(2)}, \dots, f^{(t)}$, hence the AVG in the names. The algorithms as well as the proofs of their corresponding theorems can be found in Appx. B.3 and B.4.

**Theorem 5.** *For any $\epsilon, \delta \in (0,1)$, $24\epsilon/25 < \alpha < 1$, and hypothesis class $\mathcal{F}$ of VC dimension $d$, Algorithm NR1-AVG returns a classifier $f_{NR1\text{-}AVG}$ such that for the expected error $\overline{err}_{D_i}(f_{NR1\text{-}AVG}) \leq (1+\alpha)\text{OPT} + \epsilon$ holds for all $i \in [k]$ with probability $1 - \delta$ using $m$ samples, where*

$$m = O\Big(\frac{\ln(k)}{\alpha^3 \epsilon^2}\Big(d\ln\Big(\frac{1}{\epsilon}\Big) + k\ln\Big(\frac{k}{\delta}\Big)\Big)\Big).$$

**Theorem 6.** *For any $\epsilon, \delta \in (0,1)$, $30\epsilon/29 < \alpha < 1$, and hypothesis class $\mathcal{F}$ of VC dimension $d$, Algorithm NR2-AVG returns a classifier $f_{NR2\text{-}AVG}$ such that for the expected error $\overline{err}_{D_i}(f_{NR2\text{-}AVG}) \leq (1+\alpha)\text{OPT} + \epsilon$ holds for all $i \in [k]$ with probability $1 - \delta$ using $m$ samples, where*

$$m = O\Big(\frac{1}{\alpha^3 \epsilon^2}\ln\Big(\frac{k}{\delta}\Big)\Big((d+k)\ln\Big(\frac{1}{\epsilon}\Big) + \ln\Big(\frac{1}{\delta}\Big)\Big)\Big).$$

## 5 Discussion

The problem has four parameters, $d$, $k$, $\epsilon$ and $\delta$, so there are many ways to compare the sample complexity of the algorithms. In the non-realizable setting there is one more parameter $\alpha$, but this is set to be a constant in the beginning of the algorithms. Our sample complexity upper bounds are summarized in the following table.

Table 1: Sample complexity upper bounds

|  | Algorithm 1 | Algorithm 2 |
|---|---|---|
| Realizable | $O\Big(\frac{\ln(k)}{\epsilon}\Big(d\ln\Big(\frac{1}{\epsilon}\Big)+k\ln\Big(\frac{k}{\delta}\Big)\Big)\Big)$ | $O\Big(\frac{\ln(k/\delta)}{\epsilon}\Big(d\ln\Big(\frac{1}{\epsilon}\Big)+k+\ln\Big(\frac{1}{\delta}\Big)\Big)\Big)$ |
| Non-realizable ($2+\alpha$ approx.) | $O\Big(\frac{\ln(k)}{\alpha^4\epsilon}\Big(d\ln\Big(\frac{1}{\epsilon}\Big)+k\ln\Big(\frac{k}{\delta}\Big)\Big)\Big)$ | $O\Big(\frac{\ln(k/\delta)}{\alpha^4\epsilon}\Big(d\ln\Big(\frac{1}{\epsilon}\Big)+k\ln\Big(\frac{1}{\alpha}\Big)+\ln\Big(\frac{1}{\delta}\Big)\Big)\Big)$ |
| Non-realizable (randomized) | $O\Big(\frac{\ln(k)}{\alpha^3\epsilon^2}\Big(d\ln\Big(\frac{1}{\epsilon}\Big)+k\ln\Big(\frac{k}{\delta}\Big)\Big)\Big)$ | $O\Big(\frac{\ln(k/\delta)}{\alpha^3\epsilon^2}\Big((d+k)\ln\Big(\frac{1}{\epsilon}\Big)+\ln\Big(\frac{1}{\delta}\Big)\Big)\Big)$ |

Usually $\delta$ can be considered constant, since it represents the required error probability, or, in the high success probability regime, $\delta = \frac{1}{poly(k)}$. For both of these natural settings, we can see that Algorithm 2 is better than Algorithm 1, except for the case of the expected error guarantee. If we assume $k = \Theta(d)$, then Algorithm 2 is always better than Algorithm 1.

In the realizable setting, Algorithm R1 is always better than the algorithm of [5] for the centralized variant of the problem and matches their number of samples in the personalized variant. In addition, Theorem 4.1 of [5] states that the sample complexity of any algorithm in the collaborative model is $\Omega\Big(\frac{k}{\epsilon}\ln\Big(\frac{k}{\delta}\Big)\Big)$, given that $d = \Theta(k)$ and $\epsilon, \delta \in (0, 0.1)$, and this holds even for the personalized variant. For $d = \Theta(k)$, the sample complexity of Algorithm R2 is exactly $\ln\Big(\frac{k}{\delta}\Big)$ times the sample complexity for learning one task. Furthermore, when $|\mathcal{F}| = 2^d$ (e.g. the hard instance for the lower bound of [5]), only $m_{\epsilon,\delta} = O\Big(\frac{1}{\epsilon}\Big(d+\ln\Big(\frac{1}{\delta}\Big)\Big)\Big)$ samples are required in the non-collaborative setting instead of the general bound of the VC theorem, so the sample complexity bound for Algorithm R2 is $O\Big(\ln\Big(\frac{k}{\delta}\Big)\frac{1}{\epsilon}\Big(d+k+\ln\Big(\frac{1}{\delta}\Big)\Big)\Big)$ and matches exactly the lower bound of [5] up to lower order terms.

In the non-realizable setting, our generalization of algorithms R1 and R2, NR1 and NR2 respectively, have the same sample complexity as in the realizable setting and match the error guarantee for $\text{OPT} = 0$. If $\text{OPT} \neq 0$, they guarantee an error of a factor 2 multiplicatively on $\text{OPT}$. The randomized classifiers returned by Algorithms NR1-AVG and NR2-AVG avoid this factor of 2 in their expected error guarantee. However, to learn such classifiers, there are required $O\Big(\frac{1}{\epsilon}\Big)$ times more samples.

## Acknowledgements

We thank the anonymous reviewers for their helpful remarks and for pointing us to the idea of slightly modifying the algorithms in the non-realizable setting so that the optimal error is unknown. This work was partially supported by NSF CAREER 1750716 and a Graduate fellowship from Northeastern University's College of Computer and Information Science.

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
