[Supplementary Material]

# A Proofs of Section 3

## A.1 Proof of Theorem 1

To prove the correctness and sample complexity of Algorithm R1, we need to prove Lemma A.2, which describes the set $G_r$ that the TEST returns. This proof uses the following multiplicative forms of the Chernoff bounds (proved as in Theorems 4.4 and 4.5 of [12]).

**Lemma A.1** (Chernoff Bounds). *If $X$ is the average of $n$ independent random variables taking values in $\{0,1\}$, then*

$$\Pr[X \leq (1-s)\,\mathbb{E}[X]] \leq \exp\Big(-\frac{s^2\,\mathbb{E}[X]n}{2}\Big), \tag{1}$$

$$\Pr[X \geq (1+s)\,\mathbb{E}[X]] \leq \exp\Big(-\frac{s^2\,\mathbb{E}[X]n}{3}\Big), \tag{2}$$

$$\Pr[X \geq (1+s)\,\mathbb{E}[X]] \leq \exp\Big(-\frac{s\,\mathbb{E}[X]n}{3}\Big), \tag{3}$$

*where the latter inequality holds for $s \geq 1$ and the first two hold for $s \in (0,1)$.*

**Lemma A.2.** TEST$(f^{(r)}, k, \epsilon', \delta')$ *is such that the following two properties hold, each with probability at least $1-\delta'$, for all $i \in [k]$ and for a given round $r \in [t]$.*

  (a) *If $err_{D_i}(f^{(r)}) > \epsilon'$, then $i \notin G_r$.*

  (b) *If $err_{D_i}(f^{(r)}) \leq \frac{\epsilon'}{2}$, then $i \in G_r$.*

*Proof of Lemma A.2.* For this proof we assume that the number of samples $|T_i|$ for each $i \in [k]$ must be at least $\frac{32}{\epsilon'}\ln\big(\frac{k}{\delta'}\big) = O\big(\frac{1}{\epsilon'}\ln\big(\frac{k}{\delta'}\big)\big)$. For a given round $r \in [t]$:

  (a) Assume $err_{D_i}(f^{(r)}) > \epsilon'$ for some $i \in [k]$. Then

$$\Pr\Big[i \in G_r\Big]$$
$$= \Pr\Big[err_{T_i}(f^{(r)}) \leq \tfrac{3}{4}\epsilon'\Big]$$
$$< \Pr\Big[err_{T_i}(f^{(r)}) \leq \big(1-\tfrac{1}{4}\big)err_{D_i}(f^{(r)})\Big]$$
$$\overset{(1)}{\leq} \exp\Big(-\tfrac{1}{2}\big(\tfrac{1}{4}\big)^2 err_{D_i}(f^{(r)})|T_i|\Big)$$
$$< \exp\Big(-\tfrac{1}{32}\epsilon'|T_i|\Big)$$
$$\leq \exp\Big(-\tfrac{1}{32}\epsilon'\tfrac{32}{\epsilon'}\ln\big(\tfrac{k}{\delta'}\big)\Big)$$
$$\leq \tfrac{\delta'}{k}.$$

    Hence, by union bound, $err_{D_i}(f^{(r)}) > \epsilon' \Rightarrow i \notin G_r$ holds for all $i \in [k]$ with probability at least $1-\delta'$.

  (b) Assume $err_{D_i}(f^{(r)}) \leq \frac{\epsilon'}{2}$ for some $i \in [k]$. We consider two cases and we apply the Chernoff bounds with $s = \frac{\epsilon'}{4err_{D_i}(f^{(r)})}$. Note that if $err_{D_i}(f^{(r)}) = 0$ then $err_{T_i}(f^{(r)}) = 0$ and the property holds. So we only need to consider $err_{D_i}(f^{(r)}) \neq 0$. First, we need to prove that

$$\tfrac{3\epsilon'}{4} \geq (1+s)err_{D_i}(f^{(r)})$$
$$\Leftrightarrow \tfrac{3\epsilon'}{4err_{D_i}(f^{(r)})} \geq 1 + \tfrac{\epsilon'}{4err_{D_i}(f^{(r)})}$$
$$\Leftrightarrow \tfrac{\epsilon'}{2err_{D_i}(f^{(r)})} \geq 1,$$

    which is true.

*Case 1.* If $\text{err}_{D_i}(f^{(r)}) > \frac{\epsilon'}{4}$, which implies $s < 1$, then

$$\Pr\left[i \notin G_r\right]$$
$$= \Pr\left[\text{err}_{T_i}(f^{(r)}) > \frac{3}{4}\epsilon'\right]$$
$$\leq \Pr\left[\text{err}_{T_i}(f^{(r)}) \geq \left(1 + s\right)\text{err}_{D_i}(f^{(r)})\right]$$
$$\overset{(2)}{\leq} \exp\left(-\frac{1}{3}\left(\frac{\epsilon'}{4\text{err}_{D_i}(f^{(r)})}\right)^2 \text{err}_{D_i}(f^{(r)})|T_i|\right)$$
$$= \exp\left(-\frac{\epsilon'^2}{48\text{err}_{D_i}(f^{(r)})}|T_i|\right)$$
$$\leq \exp\left(-\frac{1}{48}2\epsilon'\frac{24}{\epsilon'}\ln\left(\frac{k}{\delta'}\right)\right)$$
$$\leq \frac{\delta'}{k}.$$

*Case 2.* If $\text{err}_{D_i}(f^{(r)}) \leq \frac{\epsilon'}{4}$, which implies $s \geq 1$, then:

$$\Pr\left[i \notin G_r\right]$$
$$= \Pr\left[\text{err}_{T_i}(f^{(r)}) > \frac{3}{4}\epsilon'\right]$$
$$\leq \Pr\left[\text{err}_{T_i}(f^{(r)}) \geq \left(1 + s\right)\text{err}_{D_i}(f^{(r)})\right]$$
$$\overset{(3)}{\leq} \exp\left(-\frac{1}{3}\frac{\epsilon'}{4\text{err}_{D_i}(f^{(r)})}\text{err}_{D_i}(f^{(r)})|T_i|\right)$$
$$= \exp\left(-\frac{\epsilon'}{3}|T_i|\right)$$
$$\leq \exp\left(-\frac{\epsilon'}{3}\frac{3}{\epsilon'}\ln\left(\frac{k}{\delta'}\right)\right)$$
$$\leq \frac{\delta'}{k}.$$

Hence, by union bound, $\text{err}_{D_i}(f^{(r)}) \leq \frac{\epsilon'}{2} \Rightarrow i \in G_r$ holds for all $i \in [k]$ with probability at least $1 - \delta'$.

$\square$

*Proof of Theorem 1.* First, we prove that Algorithm R1 indeed learns a good classifier, meaning that for every player $i \in [k]$ the returned classifier $f_{\text{R1}}$ has error $\text{err}_{D_i}(f_{\text{R1}}) \leq \epsilon$ with probability at least $1 - \delta$.

Let $e_i^{(r)}$ denote the number of rounds, up until and including round $r$, that $i$ did not pass the TEST. More formally, $e_i^{(r)} = |\{r' \mid r' \in [r] \text{ and } i \notin G_{r'}\}|$.

**Claim 1.** *With probability at least* $1 - \frac{2\delta}{3}$, $e_i^{(t)} < 0.4t \ \forall i \in [k]$.

From Lemma A.2(a) and union bound, with probability at least $1 - t\delta' = 1 - \frac{\delta}{3}$, the number of functions that have error more than $\epsilon'$ on $D_i$ is the same as the number of rounds that $i$ did not pass the TEST, for all $i \in [k]$. So, if the claim holds, with probability at least $1 - (\frac{2}{3} + \frac{1}{3})\delta = 1 - \delta$, less than $0.4t$ functions have error more than $\epsilon'$ on $D_i$, for all $i \in [k]$. Equivalently, with probability at least $1 - \delta$, more than $0.6t$ functions have error at most $\epsilon'$ on $D_i$, for all $i \in [k]$. As a result, with probability at least $1 - \delta$, the error of the majority of the functions is $\text{err}_{D_i}(f_{\text{R1}}) \leq \frac{0.6}{0.1}\epsilon' = \epsilon$ for all $i \in [k]$.

Let us now prove the claim.

*Proof of Claim 1.* Recall that $\Phi^{(r)} = \sum_{i=1}^{k} w_i^{(r)}$ is the potential function in round $r$. By linearity of expectation, the following holds for the error on the mixture of distributions:

$$
\begin{aligned}
\text{err}_{\tilde{D}^{(r-1)}}(f^{(r)}) &= \frac{1}{\Phi^{(r-1)}}\sum_{i=1}^{k}\left(w_i^{(r-1)}\text{err}_{D_i}(f^{(r)})\right) \\
&\geq \frac{1}{\Phi^{(r-1)}}\sum_{i \notin G_r}\left(w_i^{(r-1)}\text{err}_{D_i}(f^{(r)})\right)
\end{aligned}
\tag{4}
$$

From the VC theorem, it holds that, since $f^{(r)} = \mathcal{O}_{\mathcal{F}}(S^{(r)})$ and $|S^{(r)}| = m_{\epsilon'/16,\delta'}$, with probability at least $1 - \delta'$, $\text{err}_{\tilde{D}^{(r-1)}}(f^{(r)}) \leq \frac{\epsilon'}{16}$. From Lemma A.2(b), with probability at least $1 - \delta'$, $\text{err}_{D_i}(f^{(r)}) \geq \frac{\epsilon'}{2}$ for all

$i \notin G_r$. So with probability at least $1 - 2\delta'$ the two hold simultaneously. Combining these inequalities with (4), we get that with probability at least $1 - 2\delta'$, $\frac{\epsilon'}{16} \geq \frac{1}{\Phi^{(r-1)}} \sum_{i=1}^{k} \left( w_i^{(r-1)} \frac{\epsilon'}{2} \right) \Leftrightarrow \sum_{i \notin G_r} w_i^{(r-1)} \leq \frac{1}{8} \Phi^{(r-1)}$.

Since only the weights of players $i \notin G_r$ are doubled, it holds that for a given round $r$

$$\Phi^{(r)} \leq \Phi^{(r-1)} + \sum_{i \notin G_r} w_i^{(r-1)} \leq \frac{9}{8} \Phi^{(r-1)}.$$

Therefore with probability at least $1 - 2t\delta' = 1 - \frac{2\delta}{3}$, the inequality holds for all rounds, by union bound. By induction:

$$\Phi^{(t)} \leq \left( \frac{9}{8} \right)^t \Phi^{(0)} = \left( \frac{9}{8} \right)^t k$$

Also, for every $i \in [k]$ it holds that $w_i^{(t)} = 2^{e_i^{(t)}}$, as each weight is only doubled every time $i$ does not pass the TEST. Since the potential function is the sum of all weights, the following inequality is true.

$$w_i^{(t)} \leq \Phi^{(t)}$$
$$\Rightarrow 2^{e_i^{(t)}} \leq \left( \frac{9}{8} \right)^t k$$
$$\Rightarrow e_i^{(t)} \leq t \log \left( \frac{9}{8} \right) + \log(k)$$
$$\Rightarrow e_i^{(t)} \leq 0.17t + 0.2t < 0.4t$$

So with probability at least $1 - \frac{2\delta}{3}$, $e_i^{(t)} < 0.4t \ \forall i \in [k]$. ∎

As for the total number of samples, it is the sum of TEST's samples and the $m_{\epsilon'/16,\delta'}$ samples for each round. Since TEST is called $t = 5\lceil \log(k) \rceil$ times and each time requests $O\left( \frac{1}{\epsilon'} \ln \left( \frac{k}{\delta'} \right) \right)$ samples from each of the $k$ players, the total number of samples that it requests is $O\left( \log(k) \frac{k}{\epsilon'} \ln \left( \frac{k}{\delta'} \right) \right)$. Substituting $\epsilon' = \epsilon/6$ and $\delta' = \delta/(3t) = \delta/(15\lceil \log(k) \rceil)$, this yields

$$O\left( \frac{\log(k)}{\epsilon} k \ln \left( \frac{k \log(k)}{\delta} \right) \right) = O\left( \frac{\log(k)}{\epsilon} k \ln \left( \frac{k}{\delta} \right) \right)$$

samples in total.

In addition, the sum of the $m_{\epsilon'/16,\delta'}$ samples drawn in each round to learn the classifier for the mixture for $t = 5\lceil \log(k) \rceil$ rounds is $O\left( \frac{\log(k)}{\epsilon'} \left( d \ln \left( \frac{1}{\epsilon'} \right) + \ln \left( \frac{1}{\delta'} \right) \right) \right)$. Again, substituting $\epsilon'$ and $\delta'$, we get:

$$O\left( \frac{\log(k)}{\epsilon} \left( d \ln \left( \frac{1}{\epsilon} \right) + \ln \left( \frac{\log(k)}{\delta} \right) \right) \right)$$

samples in total.

Hence, the overall bound is:

$$O\left( \frac{\log(k)}{\epsilon} \left( d \ln \left( \frac{1}{\epsilon} \right) + k \ln \left( \frac{k}{\delta} \right) \right) \right)$$

□

## A.2   Proof of Lemma 2.1

*Proof of Lemma 2.1.* For this proof, we assume that the number of samples $|T_i|$ for each $i \in [k]$ must be at least $\frac{148}{\epsilon'} = O\left( \frac{1}{\epsilon'} \right)$. For given $r \in [t]$ and $i \in [k]$:

(a) Assume $\text{err}_{D_i}(f^{(r)}) > \epsilon'$. Then

$$\Pr \left[ i \in G_r \right]$$
$$= \Pr \left[ \text{err}_{T_i}(f^{(r)}) \leq \frac{3}{4} \epsilon' \right]$$

$$< \Pr\left[\mathrm{err}_{T_i}(f^{(r)}) \leq \left(1 - \tfrac{1}{4}\right)\mathrm{err}_{D_i}(f^{(r)})\right]$$

$$\overset{(1)}{\leq} \exp\left(-\tfrac{1}{2}\left(\tfrac{1}{4}\right)^2 \mathrm{err}_{D_i}(f^{(r)})|T_i|\right)$$

$$< \exp\left(-\tfrac{1}{32}\epsilon'|T_i|\right)$$

$$\leq \exp\left(-\tfrac{1}{32}\epsilon'\tfrac{148}{\epsilon'}\right)$$

$$< 0.01.$$

Hence, $\mathrm{err}_{D_i}(f^{(r)}) > \epsilon' \Rightarrow i \notin G_r$ holds with probability at least 0.99.

(b) Assume $\mathrm{err}_{D_i}(f^{(r)}) \leq \tfrac{\epsilon'}{2}$. We consider two cases and we apply the Chernoff bounds with $s = \frac{\epsilon'}{4\mathrm{err}_{D_i}(f^{(r)})}$.

Note that if $\mathrm{err}_{D_i}(f^{(r)}) = 0$ then $\mathrm{err}_{T_i}(f^{(r)}) = 0$ and the property holds. So we only need to consider $\mathrm{err}_{D_i}(f^{(r)}) \neq 0$. First, we need to prove that

$$\tfrac{3\epsilon'}{4} \geq (1+s)\mathrm{err}_{D_i}(f^{(r)})$$

$$\Leftrightarrow \tfrac{3\epsilon'}{4\mathrm{err}_{D_i}(f^{(r)})} \geq 1 + \tfrac{\epsilon'}{4\mathrm{err}_{D_i}(f^{(r)})}$$

$$\Leftrightarrow \tfrac{\epsilon'}{2\mathrm{err}_{D_i}(f^{(r)})} \geq 1,$$

which is true.

*Case 1.* If $\mathrm{err}_{D_i}(f^{(r)}) > \tfrac{\epsilon'}{4}$, which implies $s < 1$, then

$$\Pr\left[i \notin G_r\right]$$

$$= \Pr\left[\mathrm{err}_{T_i}(f^{(r)}) > \tfrac{3}{4}\epsilon'\right]$$

$$\leq \Pr\left[\mathrm{err}_{T_i}(f^{(r)}) \geq \left(1+s\right)\mathrm{err}_{D_i}(f^{(r)})\right]$$

$$\overset{(2)}{\leq} \exp\left(-\tfrac{1}{3}\left(\tfrac{\epsilon'}{4\mathrm{err}_{D_i}(f^{(r)})}\right)^2 \mathrm{err}_{D_i}(f^{(r)})|T_i|\right)$$

$$= \exp\left(-\tfrac{\epsilon'^2}{48\mathrm{err}_{D_i}(f^{(r)})}|T_i|\right)$$

$$\leq \exp\left(-\tfrac{1}{48}2\epsilon'\tfrac{148}{\epsilon'}\right)$$

$$< 0.01.$$

*Case 2.* If $\mathrm{err}_{D_i}(f^{(r)}) \leq \tfrac{\epsilon'}{4}$, which implies $s \geq 1$, then

$$\Pr\left[i \notin G_r\right]$$

$$= \Pr\left[\mathrm{err}_{T_i}(f^{(r)}) > \tfrac{3}{4}\epsilon'\right]$$

$$\leq \Pr\left[\mathrm{err}_{T_i}(f^{(r)}) \geq \left(1+s\right)\mathrm{err}_{D_i}(f^{(r)})\right]$$

$$\overset{(3)}{\leq} \exp\left(-\tfrac{1}{3}\tfrac{\epsilon'}{4\mathrm{err}_{D_i}(f^{(r)})}\mathrm{err}_{D_i}(f^{(r)})|T_i|\right)$$

$$= \exp\left(-\tfrac{\epsilon'}{12}|T_i|\right)$$

$$\leq \exp\left(-\tfrac{\epsilon'}{12}\tfrac{148}{\epsilon'}\right)$$

$$< 0.01.$$

Hence, $\mathrm{err}_{D_i}(f^{(r)}) \leq \tfrac{\epsilon'}{2} \Rightarrow i \in G_r$ holds with probability at least 0.99.

$\square$

# B  Algorithms and proofs of Section 4

## B.1  Algorithm NR1

---
**Algorithm** NR1

---
1: **Initialization**: $\forall i \in [k]$ $w_i^{(0)} := 1$; $\alpha' := \alpha/35$; $t := 2\lceil \ln(k)/\alpha'^3 \rceil$; $\epsilon' := \epsilon/60$; $\delta' := \delta/(4t)$;

2: **for** $r = 1, \ldots, t$ **do**

3:  $\tilde{D}^{(r-1)} \leftarrow \frac{1}{\Phi^{(r-1)}} \sum_{i=1}^{k} \left( w_i^{(r-1)} D_i \right)$, where $\Phi^{(r-1)} := \sum_{i=1}^{k} w_i^{(r-1)}$;

4:  Draw a sample set $S^{(r)}$ of size $O\left( \frac{1}{\alpha'\epsilon'} \left( d \ln \left( \frac{1}{\epsilon'} \right) + \ln \left( \frac{1}{\delta'} \right) \right) \right)$ from $\tilde{D}^{(r-1)}$;

5:  $f^{(r)} \leftarrow \mathcal{O}_{\mathcal{F}}(S^{(r)})$;

6:  **for** $i = 1, \ldots, k$ **do**

7:   Draw a sample set $T_i$ of size $O\left( \frac{1}{\alpha'\epsilon'} \ln \left( \frac{k}{\delta'} \right) \right)$ from $D_i$;

8:   $s_i^{(r)} \leftarrow \min \left( \frac{\mathrm{err}_{T_i}(f^{(r)})\alpha'^2}{(1+3\alpha')\mathrm{err}_{S^{(r)}}(f^{(r)})+3\epsilon'}, \alpha' \right)$

9:   **Update**: $w_i^{(r)} \leftarrow w_i^{(r-1)}(1 + s_i^{(r)})$

10:  **end for**

11: **end for**

12:

13: **return**  $f_{\mathrm{NR1}} = \mathrm{maj}(\{f^{(r)}\}_{r=1}^{t})$;

---

The algorithms of this section share many useful properties. We will first prove some of these properties and then prove each one of the Theorems 3, 4, 5, and 6.

**Corollary** (of Lemma A.1)**.** *If $X$ is the average of $n$ independent random variables taking values in $\{0,1\}$, then:*

$$\Pr[X \leq (1-\alpha)\, \mathbb{E}[X] - \epsilon] \leq \exp(-\alpha\epsilon n) \;\; \forall \alpha, \epsilon \in (0,1) \tag{5}$$

$$\Pr[X \geq (1+\alpha)\, \mathbb{E}[X] + \epsilon] \leq \exp\left( -\frac{\alpha\epsilon n}{3} \right) \;\; \forall \alpha, \epsilon \in (0,1) \tag{6}$$

*Proof.* We first prove inequality (5). Note that if $\mathbb{E}[X] \leq \epsilon$ then the inequality is trivially true so we only need to consider $\mathbb{E}[X] > \epsilon$. Let $s = \alpha + \frac{\epsilon}{\mathbb{E}[X]}$. Notice that $s^2 \geq \frac{2\alpha\epsilon}{\mathbb{E}[X]}$. Thus, by inequality (1),

$$\Pr[X \leq (1-\alpha)\, \mathbb{E}[X] - \epsilon] \leq \exp(-s^2\, \mathbb{E}[X]n/2) \leq \exp(-\alpha\epsilon n).$$

Next we prove inequality (6). Again let $s = \alpha + \frac{\epsilon}{\mathbb{E}[X]}$. If $s < 1$ then by inequality (2,

$$\Pr[X \geq (1+\alpha)\, \mathbb{E}[X] + \epsilon] \leq \exp(-s^2\, \mathbb{E}[X]n/3) \leq \exp(-2\alpha\epsilon n/3).$$

If $s \geq 1$ then by inequality (3),

$$\Pr[X \geq (1+\alpha)\, \mathbb{E}[X] + \epsilon] \leq \exp(-s\, \mathbb{E}[X]n/3) \leq \exp(-\epsilon n/3) \leq \exp(-\alpha\epsilon n/3).$$

$\square$

Lemma B.1 proves that the error of the classifier $f^{(r)}$ of each round on the weighted mixture of distributions is low. It holds due to a known extension of the VC Theorem and Chernoff bounds, but we prove it here for our parameters for completeness.

**Lemma B.1.** *With probability at least $1 - \delta/2$, for all rounds $r \in [t]$:*

(a) $(1 + 3\alpha')err_{S^{(r)}}(f^{(r)}) + 3\epsilon' \leq (1 + 7\alpha')\texttt{OPT} + 19\epsilon'$.

(b) $err_{\tilde{D}^{(r-1)}}(f^{(r)}) \leq (1 + \alpha')err_{S^{(r)}}(f^{(r)}) + \epsilon'$.

*Proof.* Let $S^{(r)}$ be a set of samples of size $C \cdot \frac{1}{\alpha'\epsilon'}\left(d\ln\left(\frac{1}{\epsilon'}\right) + \ln\left(\frac{1}{\delta'}\right)\right)$ drawn from $\tilde{D}^{(r-1)}$, where $C$ is a constant. We will prove that for large enough constant $C$ the two statements hold simultaneously for all rounds, each with probability at least $1 - t\delta'$. It suffices to prove that each statement in each round holds with probability at least $1 - \delta'$. For a given round $r$:

(a) By $f^*$'s definition it holds that $err_{D_i}(f^*) \leq \texttt{OPT} + \epsilon' \; \forall i \in [k]$, so it must also hold that $err_{\tilde{D}^{(r-1)}}(f^*) \leq \texttt{OPT} + \epsilon'$, since $\tilde{D}^{(r-1)}$ is a weighted average of the distributions. From the Corollary it holds that $\Pr[err_{S^{(r)}}(f^*) \geq (1 + \alpha')err_{\tilde{D}^{(r-1)}}(f^*) + \epsilon'] \leq \exp(-\alpha'\epsilon'|S^{(r)}|/3) \leq \delta'$ and since $\alpha' \leq 1$, it is easy to see that with probability at least $1 - \delta'$,

$$err_{S^{(r)}}(f^*) \leq (1 + \alpha')\texttt{OPT} + 3\epsilon' \qquad (7)$$

Since $f^{(r)}$ is the error minimizing classifier for the sample $S^{(r)}$, it holds that $err_{S^{(r)}}(f^{(r)}) \leq err_{S^{(r)}}(f^*) + \epsilon'$. Therefore,

$$(1 + 3\alpha')err_{S^{(r)}}(f^{(r)}) + 3\epsilon' \leq (1 + 3\alpha')err_{S^{(r)}}(f^*) + 7\epsilon' \overset{(7)}{\leq} (1 + 7\alpha')\texttt{OPT} + 19\epsilon'.$$

(b) We prove the second statement for all $f \in \mathcal{F}$, using Theorem 5.7 from [1]. The theorem states that for every $h \in \mathcal{H}$, it holds that $err_D(h) \leq (1 + \gamma)err_S(h) + \beta$ with probability at least $1 - 4\Pi_{\mathcal{H}}(2m)\exp\left(\frac{-\gamma\beta m}{4(\gamma+1)}\right)$, where $S$ is a sample of size $m$ drawn from a distribution $D$ on $\mathcal{X} \times \{0,1\}$, $\gamma > 2\beta$, and $\Pi_{\mathcal{H}}(n) = \max\{|\mathcal{H}_{|S}| : S \subseteq \mathcal{X} \text{ and } |S| = n\}$ is the growth function of $\mathcal{H}$.

We apply Theorem 5.7 for $\gamma = \alpha'$, $\beta = \epsilon'$, $D = \tilde{D}^{(r-1)}$, $S = S^{(r)}$, $\mathcal{H} = \mathcal{F}$. Since the VC-dimension of $\mathcal{F}$ is $d$, from [[1], Theorem 3.7] it holds that $\Pi_{\mathcal{F}}(2m) \leq \left(\frac{2em}{d}\right)^d$. In our setting, the theorem states that, given round $r$, for every $f \in \mathcal{F}$, it holds that $err_{\tilde{D}^{(r-1)}}(f) \leq (1 + \alpha')err_{S^{(r)}}(f) + \epsilon'$ with probability at least $1 - 4\left(\frac{2em}{d}\right)^d\exp\left(\frac{-\alpha'\epsilon'm}{4(\alpha'+1)}\right)$. It remains to prove that, for large enough $C$, $m = C \cdot \frac{1}{\alpha'\epsilon'}\left(d\ln\left(\frac{1}{\epsilon'}\right) + \ln\left(\frac{1}{\delta'}\right)\right)$ samples suffice to guarantee that $4\left(\frac{2em}{d}\right)^d\exp\left(\frac{-\alpha'\epsilon'm}{4(\alpha'+1)}\right) \leq \delta'$ so that the statement holds with probability at least $1 - \delta'$. It suffices to prove that for the given $m$:

$\ln(4) + d\ln(2e) + d\ln\left(\frac{m}{d}\right) - \frac{\alpha'}{8}\epsilon'm \leq -\ln\left(\frac{1}{\delta'}\right)$

$\Leftrightarrow \ln(4) + d\ln(2e) + d\ln\left(\frac{m}{d}\right) + \ln\left(\frac{1}{\delta'}\right) \leq \frac{C}{8}d\ln\left(\frac{1}{\epsilon'}\right) + \frac{C}{8}\ln\left(\frac{1}{\delta'}\right)$.

We consider two cases:

   i. If $d\ln\left(\frac{1}{\epsilon'}\right) \geq \ln\left(\frac{1}{\delta'}\right)$, then $\frac{m}{d} \leq \frac{2C}{\alpha'\epsilon'}\ln\left(\frac{1}{\epsilon'}\right) < \frac{C}{\epsilon'^2}\ln\left(\frac{1}{\epsilon'}\right)$. So to prove the statement, it suffices to prove that

$$\ln(4) + d\ln(2e) + d\left(\ln(C) + 2\ln\left(\frac{1}{\epsilon'}\right) + \ln\ln\left(\frac{1}{\epsilon'}\right)\right) + \ln\left(\frac{1}{\delta'}\right) \leq \frac{C}{8}d\ln\left(\frac{1}{\epsilon'}\right) + \frac{C}{8}\ln\left(\frac{1}{\delta'}\right).$$

    The latter inequality holds for large enough $C$.

   ii. If $d\ln\left(\frac{1}{\epsilon'}\right) \leq \ln\left(\frac{1}{\delta'}\right)$, then $\frac{m}{d} \leq \frac{2C}{\alpha'\epsilon'}\frac{\ln(1/\delta')}{d} < \frac{C}{\epsilon'^2}\frac{\ln(1/\delta')}{d}$. So to prove the statement, it suffices to prove that

$$\ln(4) + d\ln(2e) + d\left(\ln(C) + 2\ln\left(\frac{1}{\epsilon'}\right) + \ln\left(\frac{\ln(1/\delta')}{d}\right)\right) + \ln\left(\frac{1}{\delta'}\right) \leq \frac{C}{8}d\ln\left(\frac{1}{\epsilon'}\right) + \frac{C}{8}\ln\left(\frac{1}{\delta'}\right).$$

If we prove that $d \ln \left( \frac{\ln(1/\delta')}{d} \right) \leq \ln(1/\delta')$, then the inequality holds for large enough $C$. Indeed, it holds that $\ln \left( \frac{\ln(1/\delta')}{d} \right) / \frac{\ln(1/\delta')}{d} \leq \frac{1}{e}$, since $\max_{x \in \mathbb{R}} \{ \ln(x)/x \} = \frac{1}{e}$.

Thus the second statement holds too with probability at least $1 - \delta'$.

$\square$

Lemmas B.2 and B.3 give us two inequalities that are useful for all the proofs of Section 4.

**Lemma B.2.** *Let* $L_r = \{ i \in [k] \mid |err_{T_i}(f^{(r)}) - err_{D_i}(f^{(r)})| \leq \alpha' \cdot err_{D_i}(f^{(r)}) + \epsilon' \}$. *With probability* $1 - \delta/2$, *it holds that*

$$\sum_{i \in L_r} \left( w_i^{(r-1)} err_{T_i}(f^{(r)}) \right) \leq [(1 + 3\alpha')err_{S^{(r)}}(f^{(r)}) + 3\epsilon']\Phi^{(r-1)} \leq [(1 + 7\alpha')OPT + 19\epsilon']\Phi^{(r-1)}.$$

*Proof.* By linearity of expectation,

$$\begin{aligned}
err_{\tilde{D}^{(r-1)}}(f^{(r)}) &= \frac{1}{\Phi^{(r-1)}} \sum_{i=1}^{k} \left( w_i^{(r-1)} err_{D_i}(f^{(r)}) \right) \\
&\geq \frac{1}{\Phi^{(r-1)}} \sum_{i \in L_r} \left( w_i^{(r-1)} err_{D_i}(f^{(r)}) \right) \\
&\geq \frac{1}{(1 + \alpha')\Phi^{(r-1)}} \sum_{i \in L_r} \left( w_i^{(r-1)} err_{T_i}(f^{(r)}) \right) - \frac{\epsilon'}{1 + \alpha'}.
\end{aligned}$$

Therefore, $\sum_{i \in L_r} \left( w_i^{(r-1)} err_{T_i}(f^{(r)}) \right) \leq [(1 + \alpha')err_{\tilde{D}^{(r-1)}}(f^{(r)}) + \epsilon']\Phi^{(r-1)}$. By Lemma B.1(b), it follows that with probability $1 - \delta/2$,

$$\begin{aligned}
\sum_{i \in L_r} \left( w_i^{(r-1)} err_{T_i}(f^{(r)}) \right) &\leq [(1 + \alpha')(1 + \alpha')err_{S^{(r)}}(f^{(r)}) + (1 + \alpha')\epsilon' + \epsilon']\Phi^{(r-1)} \\
&\leq [(1 + 3\alpha')err_{S^{(r)}}(f^{(r)}) + 3\epsilon']\Phi^{(r-1)} \\
&\overset{\text{Lemma B.1(a)}}{\leq} [(1 + 7\alpha')OPT + 19\epsilon']\Phi^{(r-1)}.
\end{aligned}$$

$\square$

**Lemma B.3.** *For all* $i \in [k]$ *it holds that*

$$\sum_{r=1}^{t} s_i^{(r)} \leq \frac{\ln(\Phi^{(t)})}{1 - \alpha'/2}.$$

*Proof.* In every round $r$, $w_i^{(r)} = w_i^{(r-1)}(1 + s_i^{(r)})$. Therefore for any $i \in [k]$,

$$\begin{aligned}
w_i^{(t)} &= \prod_{r=1}^{t} (1 + s_i^{(r)}) \\
&\geq \prod_{r=1}^{t} \exp(s_i^{(r)} - (s_i^{(r)})^2/2) \\
&\overset{s_i^{(r)} \leq \alpha'}{\geq} \exp \left( (1 - \alpha'/2) \sum_{r=1}^{t} s_i^{(r)} \right),
\end{aligned}$$

where the second to last inequality holds since $(1 + x) \geq \exp(x - x^2/2)$ for $x \in \mathbb{R}_+$. The inequality follows since $w_i^{(t)} \leq \Phi^{(t)}$ for all $i \in [k]$.

$\square$

We will now give the proof of Theorem 3.

*Proof of Theorem 3.* By the Corollary, for a given round $r$ and player $i$,

$$\Pr[|\mathrm{err}_{T_i}(f^{(r)}) - \mathrm{err}_{D_i}(f^{(r)})| \geq \alpha' \cdot \mathrm{err}_{D_i}(f^{(r)}) + \epsilon'] \leq 2\exp(-\alpha'\epsilon'|T_i|/3).$$

If $|T_i| = \frac{3}{\epsilon'\alpha'}\ln\left(\frac{k}{\delta'}\right) = O\left(\frac{1}{\epsilon'\alpha'}\ln\left(\frac{k}{\delta'}\right)\right)$, the inequality

$$|\mathrm{err}_{T_i}(f^{(r)}) - \mathrm{err}_{D_i}(f^{(r)})| \leq \alpha' \cdot \mathrm{err}_{D_i}(f^{(r)}) + \epsilon' \tag{8}$$

holds with probability at least $1 - 2\delta'/k$. By union bound, it follows that (8) holds for every $i$ and every $r$ with probability at least $1 - 2\delta't = 1 - \delta/2$.

With probability at least $1 - \delta$ inequality (8) and the inequality of Lemma B.2 hold for all rounds and players. We restrict the rest of the proof to this event. It holds that,

$$\Phi^{(r)} = \Phi^{(r-1)} + \sum_{i=1}^{k}\left(w_i^{(r-1)} \cdot s_i^{(r)}\right)$$

$$\leq \Phi^{(r-1)} + \frac{\alpha'^2}{(1+3\alpha')\mathrm{err}_{S^{(r)}}(f^{(r)}) + 3\epsilon'}\sum_{i=1}^{k}\left(w_i^{(r-1)}\mathrm{err}_{T_i}(f^{(r)})\right)$$

$$\overset{L_r=[k]}{\leq} \Phi^{(r-1)}\left(1 + \frac{\alpha'^2}{(1+3\alpha')\mathrm{err}_{S^{(r)}}(f^{(r)}) + 3\epsilon'}[(1+3\alpha')\mathrm{err}_{S^{(r)}}(f^{(r)}) + 3\epsilon']\right)$$

$$= \Phi^{(r-1)}(1 + \alpha'^2)$$

By induction, $\Phi^{(t)} \leq \Phi^{(0)}(1+\alpha'^2)^t = k(1+\alpha'^2)^t \leq k\exp(t\alpha'^2)$. From Lemma B.3 and $t = 2\lceil\ln(k)/\alpha'^3\rceil$, it follows that

$$\sum_{r=1}^{t} s_i^{(r)} \leq \frac{\ln(k) + t\alpha'^2}{1 - \alpha'/2} \leq \frac{1+\alpha'}{1-\alpha'/2}t\alpha'^2. \tag{9}$$

Let $G_i$ be the set of rounds $r$ such that $s_i^{(r)} < \alpha'$. We consider these to be the "good" classifiers. Because of (9), we have $|[t]\setminus G_i| \leq \frac{1}{\alpha'}\sum_{r\in[t]\setminus G_i}\alpha' \leq \frac{1}{\alpha'}\sum_{r=1}^{t}s_i^{(r)} \leq \frac{1+\alpha'}{1-\alpha'/2}\alpha't$. For the classifiers of the rounds $r \in G_i$, it holds that

$$\sum_{r\in G_i}\frac{\mathrm{err}_{T_i}(f^{(r)})\alpha'^2}{(1+3\alpha')\mathrm{err}_{S^{(r)}}(f^{(r)}) + 3\epsilon'} = \sum_{r\in G_i}s_i^{(r)} \leq \sum_{r=1}^{t}s_i^{(r)} \overset{(9)}{\leq} \frac{1+\alpha'}{1-\alpha'/2}\alpha'^2t.$$

Thus, $\sum_{r\in G_i}\mathrm{err}_{T_i}(f^{(r)}) \overset{B.1(a)}{\leq} t\frac{1+\alpha'}{1-\alpha'/2}[(1+7\alpha')\mathtt{OPT} + 19\epsilon']$. From inequality (8), it follows that:

$$(1-\alpha')\sum_{r\in G_i}\mathrm{err}_{D_i}(f^{(r)}) - |G_i|\epsilon' \leq t\frac{1+\alpha'}{1-\alpha'/2}[(1+7\alpha')\mathtt{OPT} + 19\epsilon']$$

$$\Rightarrow \sum_{r\in G_i}\mathrm{err}_{D_i}(f^{(r)}) \leq t\frac{1+\alpha'}{(1-\alpha'/2)(1-\alpha')}[(1+7\alpha')\mathtt{OPT} + 19\epsilon'] + \frac{t\epsilon'}{1-\alpha'}$$

$$\Rightarrow \sum_{r\in G_i}\mathrm{err}_{D_i}(f^{(r)}) \leq [(1+12\alpha')\mathtt{OPT} + 25\epsilon']t,$$

which holds for $\alpha' < 1/12$.

For each example $e$ that is a mistake for $f_{\mathrm{NR1}}$, it must be a mistake for at least $t/2 - |[t]\setminus G_i|$ members of $G_i$. Thus the fraction of error of $f_{\mathrm{NR1}}$ is at most

$$\frac{\sum_{r\in G_i}\mathrm{err}_{D_i}(f^{(r)})}{t/2 - |[t]\setminus G_i|} \leq \frac{(1+12\alpha')\mathtt{OPT} + 25\epsilon'}{1/2 - (1+\alpha')\alpha'/(1-\alpha'/2)} \leq (2+35\alpha')\mathtt{OPT} + 60\epsilon'.$$

Having set $\alpha' = \alpha/35$ and $\epsilon' = \epsilon/60$ we get that $\mathrm{err}_{D_i}(f_{\mathrm{NR1}}) \leq (2+\alpha)\mathsf{OPT} + \epsilon$.

As for the total number of samples, it is the sum of $O(\frac{k}{\alpha'\epsilon'}\ln(k/\delta'))$ and $O\left(\frac{1}{\alpha'\epsilon'}\left(d\ln\left(\frac{1}{\epsilon'}\right) + \ln\left(\frac{1}{\delta'}\right)\right)\right)$ samples for each round. Because there are $O(\ln(k)/\alpha'^3)$ rounds, the total number of samples is

$$O\left(\frac{\ln(k)}{\alpha'^4\epsilon'}\left(k\ln\left(\frac{k}{\delta'}\right) + d\ln\left(\frac{1}{\epsilon'}\right)\right)\right) = O\left(\frac{\ln(k)}{\alpha^4\epsilon}\left(k\ln\left(\frac{k}{\delta}\right) + d\ln\left(\frac{1}{\epsilon}\right)\right)\right).$$

$\square$

## B.2 Algorithm NR2

*Proof of Theorem 4.* By the Corollary, for a given round $r$ and player $i$,

$$\Pr[|\mathrm{err}_{T_i}(f^{(r)}) - \mathrm{err}_{D_i}(f^{(r)})| \geq \alpha' \cdot \mathrm{err}_{D_i}(f^{(r)}) + \epsilon'] \leq 2\exp(-\alpha'\epsilon'|T_i|/3).$$

If $|T_i| = \frac{6}{\epsilon'\alpha'}\ln\left(\frac{\sqrt{2}}{\alpha'}\right) = O\left(\frac{1}{\epsilon'\alpha'}\ln\left(\frac{1}{\alpha'}\right)\right)$, then

$$\Pr[|\mathrm{err}_{T_i}(f^{(r)}) - \mathrm{err}_{D_i}(f^{(r)})| \geq \alpha' \cdot \mathrm{err}_{D_i}(f^{(r)}) + \epsilon'] \leq \alpha'^2. \tag{10}$$

Assuming that the inequality of Lemma B.2 holds, which is true with probability $1 - \delta/2$, it follows that

$\mathbb{E}[\Phi^{(r)} \mid \Phi^{(r-1)}]$

$$\leq \mathbb{E}\left[\Phi^{(r-1)} + \frac{\alpha'^2}{(1+3\alpha')\mathrm{err}_{S^{(r)}}(f^{(r)}) + 3\epsilon'}\sum_{i \in L_r}\left(w_i^{(r-1)}\mathrm{err}_{T_i}(f^{(r)})\right) + \sum_{i \notin L_r}\left(w_i^{(r-1)}s_i^{(r-1)}\right)\middle|\Phi^{(r-1)}\right]$$

$$\leq \mathbb{E}\left[\Phi^{(r-1)} + \frac{\alpha'^2}{(1+3\alpha')\mathrm{err}_{S^{(r)}}(f^{(r)}) + 3\epsilon'}[(1+3\alpha')\mathrm{err}_{S^{(r)}}(f^{(r)}) + 3\epsilon']\Phi^{(r-1)} + \alpha'\sum_{i \notin L_r}w_i^{(r-1)}\middle|\Phi^{(r-1)}\right]$$

$$\overset{(10)}{\leq} \Phi^{(r-1)}(1 + \alpha'^2 + \alpha'^3)$$

By the definition of expectation, $\mathbb{E}[\Phi^{(r)}] \leq \mathbb{E}[\Phi^{(r-1)}](1 + \alpha'^2 + \alpha'^3)$. So by induction and the fact that $\Phi^{(0)} = k$, $\mathbb{E}[\Phi^{(t)}] \leq k\exp(t\alpha'^2(1+\alpha'))$. Markov's inequality states that $\Pr[\Phi^{(t)} \geq \frac{\mathbb{E}[\Phi^{(t)}]}{\delta/4}] \leq \delta/4$. So with overall probability $1 - \delta/4 - \delta/2 = 1 - 3\delta/4$ it holds that $\Phi^{(t)} \leq \frac{4k}{\delta}\exp(t\alpha'^2(1+\alpha'))$.

From Lemma B.3 and $t = 2\lceil \ln(4k/\delta)/\alpha'^3\rceil$, it follows that

$$\sum_{r=1}^{t}s_i^{(r)} \leq \frac{\ln(4k/\delta) + t\alpha'^2(1+\alpha')}{1 - \alpha'/2} \leq \frac{(1+2\alpha')}{1-\alpha'/2}t\alpha'^2. \tag{11}$$

For $G_i = \{r \in [t] \mid s_i^{(r)} < \alpha'\}$, we have $|[t] \setminus G_i| \leq \frac{1+2\alpha'}{1-\alpha'/2}\alpha't$ because of (11).

Let $R_i = \{r \in [t] \mid |\mathrm{err}_{T_i}(f^{(r)}) - \mathrm{err}_{D_i}(f^{(r)})| \leq \alpha' \cdot \mathrm{err}_{D_i}(f^{(r)}) + \epsilon'\}$. For the classifiers of the rounds $r \in G_i \cap R_i$:

$$\sum_{r \in G_i \cap R_i} \mathrm{err}_{D_i}(f^{(r)}) \leq \sum_{r \in G_i \cap R_i} \frac{\mathrm{err}_{T_i}(f^{(r)})}{1 - \alpha'} + \frac{|G_i \cap R_i|\epsilon'}{1 - \alpha'}$$

$$\leq \sum_{r \in G_i \cap R_i} \frac{(1 + 3\alpha')\mathrm{err}_{S^{(r)}}(f^{(r)}) + 3\epsilon'}{\alpha'^2} \frac{\mathrm{err}_{T_i}(f^{(r)})\alpha'^2}{(1 - \alpha')[(1 + 3\alpha')\mathrm{err}_{S^{(r)}}(f^{(r)}) + 3\epsilon']} + \frac{t\epsilon'}{1 - \alpha'}$$

$$= \sum_{r \in G_i \cap R_i} \frac{(1 + 3\alpha')\mathrm{err}_{S^{(r)}}(f^{(r)}) + 3\epsilon'}{(1 - \alpha')\alpha'^2} s_i^{(r)} + \frac{t\epsilon'}{1 - \alpha'}$$

$$\overset{(11)}{\leq} \frac{(1 + 7\alpha')\mathtt{OPT} + 19\epsilon'}{(1 - \alpha')\alpha'^2} \frac{(1 + 2\alpha')}{1 - \alpha'/2} t\alpha'^2 + \frac{t\epsilon'}{1 - \alpha'}$$

$$\leq [(1 + 15\alpha')\mathtt{OPT} + 25\epsilon']t$$

which holds for $\alpha' < 1/15$.

We will now bound $|[t] \setminus R_i|$. For every round $r$, let $m^{(r)}$ be the indicator random variable of the set $[t] \setminus R_i$ and let $y^{(r)} = \alpha'^2$. It holds that for all rounds $r$, $|m^{(r)} - y^{(r)}| \leq 1$ and $m^{(r)}, y^{(r)} \geq 0$. In addition, from inequality (10) it follows that $\mathbb{E}[m^{(r)} - y^{(r)} \mid \sum_{r' < r} m^{(r')}, \sum_{r' < r} y^{(r')}] = \alpha'^2 - \alpha'^2 \leq 0$.

Using [[9], Lemma 10], with $\varepsilon = 1/2$ and $A = \alpha'^2$, we get that

$$\Pr\left[\sum_{r=1}^{t} m^{(r)} \geq 2\alpha'^2 t + 2\alpha'^2 t\right] \leq \exp(-\alpha'^2 t/2) \leq \delta/4k.$$

So $|[t] \setminus R_i| = \sum_{r=1}^{t} m^{(r)} \leq 4\alpha'^2 t$ for all $i$ with probability at least $1 - \delta/4$, by union bound.

For each example $e$ that is a mistake for $f_{\mathrm{NR2}}$, it must be a mistake for at least $t/2 - |[t] \setminus (G_i \cap R_i)|$ members of $G_i \cap R_i$. Thus, with probability at least $1 - \delta$, the fraction of error of $f_{\mathrm{NR2}}$ is at most

$$\frac{\sum_{r \in G_i \cap R_i} \mathrm{err}_{D_i}(f^{(r)})}{t/2 - |[t] \setminus (G_i \cap R_i)|} \leq \frac{(1 + 15\alpha')\mathtt{OPT} + 25\epsilon'}{t/2 - 4\alpha'^2 t - (1 + \alpha')\alpha' t/(1 - \alpha'/2)} \leq (2 + 40\alpha')\mathtt{OPT} + 64\epsilon'.$$

Having set $\alpha' = \alpha/40$ and $\epsilon' = \epsilon/64$ we get that $\mathrm{err}_{D_i}(f_{\mathrm{NR2}}) \leq (2 + \alpha)\mathtt{OPT} + \epsilon$.

As for the total number of samples, it is the sum of $O(\frac{k}{\alpha'\epsilon'} \ln(1/\alpha'))$ samples and $O\left(\frac{1}{\alpha'\epsilon'}\left(d \ln\left(\frac{1}{\epsilon'}\right) + \ln\left(\frac{1}{\delta'}\right)\right)\right)$ samples for each round. Because there are $O(\ln(k/\delta)/\alpha'^3)$ rounds, the total number of samples is

$$O\left(\frac{1}{\alpha^4 \epsilon} \ln\left(\frac{k}{\delta}\right)\left(k \ln\left(\frac{1}{\alpha}\right) + d \ln\left(\frac{1}{\epsilon}\right) + \ln\left(\frac{1}{\delta}\right)\right)\right).$$

$\square$

## B.3 Algorithm NR1-AVG

---

**Algorithm** NR1-AVG

---

1: **Initialization**: $\forall i \in [k]\ w_i^{(0)} := 1;\ \alpha' := \alpha/12;\ t := 2\lceil \ln(k)/(\epsilon'\alpha'^2) \rceil;\ \epsilon' := \epsilon/25;\ \delta' := \delta/(4t);$
2: **for** $r = 1, \ldots, t$ **do**
3:      $\tilde{D}^{(r-1)} \leftarrow \frac{1}{\Phi^{(r-1)}} \sum_{i=1}^{k} \left( w_i^{(r-1)} D_i \right)$, where $\Phi^{(r-1)} := \sum_{i=1}^{k} w_i^{(r-1)};$
4:      Draw a sample set $S^{(r)}$ of size $O\left( \frac{1}{\alpha'\epsilon'} \left( d \ln \left( \frac{1}{\epsilon'} \right) + \ln \left( \frac{1}{\delta'} \right) \right) \right)$ from $\tilde{D}^{(r-1)};$
5:      $f^{(r)} \leftarrow \mathcal{O}_{\mathcal{F}}(S^{(r)});$
6:      **for** $i = 1, \ldots, k$ **do**
7:          Draw a sample set $T_i$ of size $O\left( \frac{1}{\alpha'\epsilon'} \ln \left( \frac{k}{\delta'} \right) \right)$ from $D_i;$
8:          $s_i^{(r)} \leftarrow \frac{\operatorname{err}_{T_i}(f^{(r)})\epsilon'\alpha'}{(1+3\alpha')\operatorname{err}_{S^{(r)}}(f^{(r)})+3\epsilon'}$
9:          **Update**: $w_i^{(r)} \leftarrow w_i^{(r-1)}(1 + s_i^{(r)})$
10:      **end for**
11: **end for**
12:
13: **return** $f_{\text{NR1-AVG}}$, where $f_{\text{NR1-AVG}}(x) \overset{R}{\leftarrow} \{f^{(r)}(x)\}_{r=1}^{t};$

---

*Proof of Theorem 5.* The expected error of the returned classifier $f_{\text{NR1-AVG}}$ on player $i$'s distribution is $\overline{\operatorname{err}}_{D_i}(f_{\text{NR1-AVG}}) = \frac{1}{t}\sum_{r=1}^{t} \operatorname{err}_{D_i}(f^{(r)})$. We will prove that with probability at least $1-\delta$, $\overline{\operatorname{err}}_{D_i}(f_{\text{NR1-AVG}}) \leq (1+\alpha)\text{OPT} + \epsilon$ for all $i \in [k]$.

By the Corollary, for a given round $r$ and player $i$,

$$\Pr[|\operatorname{err}_{T_i}(f^{(r)}) - \operatorname{err}_{D_i}(f^{(r)})| \geq \alpha' \cdot \operatorname{err}_{D_i}(f^{(r)}) + \epsilon'] \leq 2\exp(-\alpha'\epsilon'|T_i|/3).$$

If $|T_i| = \frac{3}{\epsilon'\alpha'} \ln \left( \frac{k}{\delta'} \right) = O\left( \frac{1}{\epsilon'\alpha'} \ln \left( \frac{k}{\delta'} \right) \right)$, the inequality holds with probability at least $1 - 2\delta'/k$. By union bound, it follows that it holds for every $i$ and every $r$ with probability at least $1 - 2\delta't = 1 - \delta/2$.

With probability at least $1 - \delta$ the previous inequality as well as the inequality of Lemma B.2 hold for all rounds and players. We restrict the rest of the proof to this event.

It holds that,

$$\Phi^{(r)} = \Phi^{(r-1)} + \sum_{i=1}^{k} \left( w_i^{(r-1)} s_i^{(r)} \right)$$

$$\leq \Phi^{(r-1)} + \frac{\epsilon'\alpha'}{(1+3\alpha')\operatorname{err}_{S^{(r)}}(f^{(r)})+3\epsilon'} \sum_{i=1}^{k} \left( w_i^{(r-1)} \operatorname{err}_{T_i}(f^{(r)}) \right)$$

$$\overset{L_r=[k]}{\leq} \Phi^{(r-1)} \left( 1 + \frac{\epsilon'\alpha'}{(1+3\alpha')\operatorname{err}_{S^{(r)}}(f^{(r)})+3\epsilon'}[(1+3\alpha')\operatorname{err}_{S^{(r)}}(f^{(r)})+3\epsilon'] \right)$$

$$\leq \Phi^{(r-1)}(1 + \epsilon'\alpha')$$

By induction, $\Phi^{(t)} \leq k\exp(t\epsilon'\alpha')$. From Lemma B.3 and since $t = 2\lceil \ln(k)/(\epsilon'\alpha'^2) \rceil$, it follows that

$$\sum_{r=1}^{t} s_i^{(r)} \leq \frac{\ln(k) + t\epsilon'\alpha'}{1 - \alpha'/2} \leq \frac{1+\alpha'}{1-\alpha'/2}t\epsilon'\alpha'. \tag{12}$$

Therefore, the total error is:

$$
\begin{aligned}
\sum_{r=1}^{t} \mathrm{err}_{D_i}(f^{(r)}) &\leq \sum_{r=1}^{t} \frac{\mathrm{err}_{T_i}(f^{(r)})}{1-\alpha'} + \frac{t\epsilon'}{1-\alpha'} \\
&\leq \sum_{r=1}^{t} \frac{(1+3\alpha')\mathrm{err}_{S^{(r)}}(f^{(r)}) + 3\epsilon'}{\epsilon'\alpha'} \frac{\mathrm{err}_{T_i}(f^{(r)})\epsilon'\alpha'}{(1-\alpha')[(1+3\alpha')\mathrm{err}_{S^{(r)}}(f^{(r)}) + 3\epsilon']} + \frac{t\epsilon'}{1-\alpha'} \\
&= \sum_{r=1}^{t} \frac{(1+3\alpha')\mathrm{err}_{S^{(r)}}(f^{(r)}) + 3\epsilon'}{(1-\alpha')\epsilon'\alpha'} s_i^{(r)} + \frac{t\epsilon'}{1-\alpha'} \\
&\overset{(12)}{\leq} \frac{(1+7\alpha')\mathtt{OPT} + 19\epsilon'}{(1-\alpha')\epsilon'\alpha'} \frac{(1+\alpha')}{1-\alpha'/2} t\epsilon'\alpha' + \frac{t\epsilon'}{1-\alpha'} \\
&\leq [(1+12\alpha')\mathtt{OPT} + 25\epsilon']t \\
&= [(1+\alpha)\mathtt{OPT} + \epsilon]t,
\end{aligned}
$$

where the last inequality holds for $\alpha' < 1/12$ and we have set $\alpha' = \alpha/12$ and $\epsilon' = \epsilon/25$.

As for the total number of samples, it is the sum of $O(\frac{k}{\alpha'\epsilon'}\ln(k/\delta'))$ samples and $O\left(\frac{1}{\alpha'\epsilon'}\left(d\ln\left(\frac{1}{\epsilon'}\right) + \ln\left(\frac{1}{\delta'}\right)\right)\right)$ samples for each round. Because there are $O(\ln(k)/(\epsilon'\alpha'^2))$ rounds, the total number of samples is

$$
O\left(\frac{\ln(k)}{\alpha^3\epsilon^2}\left(k\ln\left(\frac{k}{\delta}\right) + d\ln\left(\frac{1}{\epsilon}\right)\right)\right).
$$

$\square$

## B.4 Algorithm NR2-AVG

---
**Algorithm** NR2-AVG
---

1: **Initialization**: $\forall i \in [k]$ $w_i^{(0)} := 1$; $\alpha' := \alpha/15$; $t := 2\lceil\ln(4k/\delta)/(\epsilon'\alpha'^2)\rceil$; $\epsilon' := \epsilon/29$; $\delta' := \delta/(4t)$;
2: **for** $r = 1, \ldots, t$ **do**
3:    $\tilde{D}^{(r-1)} \leftarrow \frac{1}{\Phi^{(r-1)}}\sum_{i=1}^{k}\left(w_i^{(r-1)}D_i\right)$, where $\Phi^{(r-1)} := \sum_{i=1}^{k} w_i^{(r-1)}$;
4:    Draw a sample set $S^{(r)}$ of size $O\left(\frac{1}{\alpha'\epsilon'}\left(d\ln\left(\frac{1}{\epsilon'}\right) + \ln\left(\frac{1}{\delta'}\right)\right)\right)$ from $\tilde{D}^{(r-1)}$;
5:    $f^{(r)} \leftarrow \mathcal{O}_{\mathcal{F}}(S^{(r)})$;
6:    **for** $i = 1, \ldots, k$ **do**
7:       Draw a sample set $T_i$ of size $O\left(\frac{1}{\alpha'\epsilon'}\ln\left(\frac{1}{\epsilon'}\right)\right)$ from $D_i$;
8:       $s_i^{(r)} \leftarrow \frac{\mathrm{err}_{T_i}(f^{(r)})\epsilon'\alpha'}{(1+3\alpha')\mathrm{err}_{S^{(r)}}(f^{(r)}) + 3\epsilon'}$
9:       **Update**: $w_i^{(r)} \leftarrow w_i^{(r-1)}(1 + s_i^{(r)})$
10:   **end for**
11: **end for**
12:
13: **return** $f_{\text{NR2-AVG}}$, where $f_{\text{NR2-AVG}}(x) \overset{R}{\leftarrow} \{f^{(r)}(x)\}_{r=1}^{t}$;

---

*Proof of Theorem 6.* The expected error of the returned classifier $f_{\text{NR2-AVG}}$ on player $i$'s distribution is $\overline{\mathrm{err}}_{D_i}(f_{\text{NR2-AVG}}) = \frac{1}{t}\sum_{r=1}^{t} \mathrm{err}_{D_i}(f^{(r)})$. We will prove that with probability at least $1 - \delta$, $\overline{\mathrm{err}}_{D_i}(f_{\text{NR2-AVG}}) \leq (1+\alpha)\mathtt{OPT} + \epsilon$ for all $i \in [k]$.

By the Corollary, for a given round $r$ and player $i$,

$$
\Pr[|\mathrm{err}_{T_i}(f^{(r)}) - \mathrm{err}_{D_i}(f^{(r)})| \geq \alpha' \cdot \mathrm{err}_{D_i}(f^{(r)}) + \epsilon'] \leq 2\exp(-\alpha'\epsilon'|T_i|/3).
$$

If $|T_i| = \frac{3}{\epsilon'\alpha'} \ln\left(\frac{2}{\epsilon'\alpha'}\right) \overset{\alpha' \geq 2\epsilon'}{=} O\left(\frac{1}{\epsilon'\alpha'} \ln\left(\frac{1}{\epsilon'}\right)\right)$, then

$$\Pr[|\mathrm{err}_{T_i}(f^{(r)}) - \mathrm{err}_{D_i}(f^{(r)})| \geq \alpha' \cdot \mathrm{err}_{D_i}(f^{(r)}) + \epsilon'] \leq \epsilon'\alpha'. \tag{13}$$

Assuming that the inequality of Lemma B.2 holds, which is true with probability $1 - \delta/2$, it follows that

$\mathbb{E}[\Phi^{(r)} \mid \Phi^{(r-1)}]$

$$= \mathbb{E}\left[\Phi^{(r-1)} + \frac{\epsilon'\alpha'}{(1+3\alpha')\mathrm{err}_{S^{(r)}}(f^{(r)}) + 3\epsilon'} \sum_{i \in L_r} \left(w_i^{(r-1)}\mathrm{err}_{T_i}(f^{(r)})\right) + \sum_{i \notin L_r} \left(w_i^{(r-1)}s_i^{(r-1)}\right) \middle| \Phi^{(r-1)}\right]$$

$$\leq \mathbb{E}\left[\Phi^{(r-1)} + \frac{\epsilon'\alpha'}{(1+3\alpha')\mathrm{err}_{S^{(r)}}(f^{(r)}) + 3\epsilon'}[(1+3\alpha')\mathrm{err}_{S^{(r)}}(f^{(r)}) + 3\epsilon']\Phi^{(r-1)} + \alpha' \sum_{i \notin L_r} w_i^{(r-1)} \middle| \Phi^{(r-1)}\right]$$

$$\overset{(13)}{\leq} \Phi^{(r-1)}(1 + \epsilon'\alpha' + \epsilon'\alpha'^2)$$

By the definition of expectation, $\mathbb{E}[\Phi^{(r)}] \leq \mathbb{E}[\Phi^{(r-1)}](1 + \epsilon'\alpha' + \epsilon'\alpha'^2)$. So by induction, $\mathbb{E}[\Phi^{(t)}] \leq k \exp(t\epsilon'\alpha'(1+\alpha'))$. Markov's inequality states that $\Pr[\Phi^{(t)} \geq \frac{\mathbb{E}[\Phi^{(t)}]}{\delta/4}] \leq \delta/4$. So with probability $1 - \delta/4 - \delta/2 = 1 - 3\delta/4$ it holds that $\Phi^{(t)} \leq \frac{4k}{\delta} \exp(t\epsilon'\alpha'(1+\alpha'))$.

From Lemma B.3 and $t = 2\lceil \ln(4k/\delta)/(\epsilon'\alpha'^2)\rceil$, it follows that

$$\sum_{r=1}^{t} s_i^{(r)} \leq \frac{\ln(4k/\delta) + t\epsilon'\alpha'(1+\alpha')}{1 - \alpha'/2} \leq \frac{(1+2\alpha')}{1-\alpha'/2}t\epsilon'\alpha'. \tag{14}$$

Let $R_i = \{r \in [t] \mid |\mathrm{err}_{T_i}(f^{(r)}) - \mathrm{err}_{D_i}(f^{(r)})| \leq \alpha' \cdot \mathrm{err}_{D_i}(f^{(r)}) + \epsilon'\}$. For the classifiers of the rounds $r \in R_i$:

$$\sum_{r \in R_i} \mathrm{err}_{D_i}(f^{(r)}) \leq \sum_{r \in R_i} \frac{\mathrm{err}_{T_i}(f^{(r)})}{1-\alpha'} + \frac{|R_i|\epsilon'}{1-\alpha'}$$

$$\leq \sum_{r \in R_i} \frac{(1+3\alpha')\mathrm{err}_{S^{(r)}}(f^{(r)}) + 3\epsilon'}{\epsilon'\alpha'} \frac{\mathrm{err}_{T_i}(f^{(r)})\epsilon'\alpha'}{(1-\alpha')[(1+3\alpha')\mathrm{err}_{S^{(r)}}(f^{(r)}) + 3\epsilon']} + \frac{t\epsilon'}{1-\alpha'}$$

$$= \sum_{r \in R_i} \frac{(1+3\alpha')\mathrm{err}_{S^{(r)}}(f^{(r)}) + 3\epsilon'}{(1-\alpha')\epsilon'\alpha'} s_i^{(r)} + \frac{t\epsilon'}{1-\alpha'}$$

$$\overset{(14)}{\leq} \frac{(1+7\alpha')\mathrm{OPT} + 19\epsilon'}{(1-\alpha')\epsilon'\alpha'} \frac{(1+2\alpha')}{1-\alpha'/2}t\epsilon'\alpha' + \frac{t\epsilon'}{1-\alpha'}$$

$$\leq [(1+15\alpha')\mathrm{OPT} + 25\epsilon']t$$

which holds for $\alpha' < 1/15$.

We will now bound $|[t] \setminus R_i|$. For every round $r$, let $m^{(r)}$ be the indicator random variable of the set $[t] \setminus R_i$ and let $y^{(r)} = \epsilon'\alpha'$. It holds that for all rounds $r$, $|m^{(r)} - y^{(r)}| \leq 1$ and $m^{(r)}, y^{(r)} \geq 0$. In addition, from inequality (13) it follows that $\mathbb{E}[m^{(r)} - y^{(r)} \mid \sum_{r'<r} m^{(r')}, \sum_{r'<r} y^{(r')}] = \epsilon'\alpha' - \epsilon'\alpha' \leq 0$.

Using [[9], Lemma 10], with $\varepsilon = 1/2$ and $A = \epsilon'\alpha'$, we get that

$$\Pr\left[\sum_{r=1}^{t} m^{(r)} \geq 2\epsilon'\alpha't + 2\epsilon'\alpha't\right] \leq \exp(-\epsilon'\alpha't/2) \leq \delta/4k.$$

So $|[t] \setminus R_i| = \sum_{r=1}^{t} m^{(r)} \leq 4\epsilon'\alpha't$ for all $i$ with probability at least $1 - \delta/4$.

Thus, for the expected error it holds that:

$$\frac{\sum\limits_{r=1}^{t} \mathrm{err}_{D_i}(f^{(r)})}{t} = \frac{\sum\limits_{r \in R_i} \mathrm{err}_{D_i}(f^{(r)}) + \sum\limits_{r \notin R_i} \mathrm{err}_{D_i}(f^{(r)})}{t}$$
$$\leq (1 + 15\alpha')\mathtt{OPT} + 25\epsilon' + 4\epsilon'\alpha' \leq (1 + 15\alpha')\mathtt{OPT} + 29\epsilon'.$$

Having set $\alpha' = \alpha/15$ and $\epsilon' = \epsilon/29$ we get that $\overline{\mathrm{err}}_{D_i}(f_{\text{NR2-AVG}}) \leq (1 + \alpha)\mathtt{OPT} + \epsilon$ with probability at least $1 - \delta$.

As for the total number of samples, it is the sum of $O(\frac{k}{\alpha'\epsilon'} \ln(1/\epsilon'))$ samples and $O\left(\frac{1}{\alpha'\epsilon'}\left(d \ln\left(\frac{1}{\epsilon'}\right) + \ln\left(\frac{1}{\delta'}\right)\right)\right)$ samples for each round. Because there are $O(\ln(k/\delta)/\epsilon'\alpha'^2)$ rounds, the total number of samples is

$$O\left(\frac{1}{\alpha^3\epsilon^2} \ln\left(\frac{k}{\delta}\right)\left((d + k) \ln\left(\frac{1}{\epsilon}\right) + \ln\left(\frac{1}{\delta}\right)\right)\right).$$

$\square$