[Reviews · NeurIPS 2018]

Reviewer 1



This paper continues the study of collaboration in the learning and obtains improved same complexity bounds. Background: The collaborative PAC model, introduced by Blum et al (NIPS’17), considers a setting where k players with different distributions D_is that are all consistent with some unknown function f^* want to (\epsilon, \delta)-learn a classifier for their own distribution. The question Blum et al. asks is what is the total “overhead” over the sample complexity of accomplishing one task, if the players can collaborate. As an example when players do not collaborate, the k tasks have to be performed individually leading to an overhead of O(k). Blum et al showed that in the personalized setting, where different players can use different classifiers, the overhead is O(log(k)) with k = O(d). They also consider minor extensions of their setting to the non-realizable setting where f^* has a small but non-zero error on the distributions. Summary: This paper obtains an improved overhead for the centralized setting of Blum et al. Their main result shows that they can get an overhead of O(log(k)) in the centralized setting. More precisely, the sample complexity they obtain is: O(ln(k/\delta)/\epsilon * (d ln(1/\epsilon) + k + ln(1/\delta) )). Their second contribution is to extend these results to the non-realizable case. Showing that if the best function f* has error \leq OPT on all distributions then there is an algorithm that returns a single function that obtains (1 + \alpha)OPT + \epsilon on all distributions using a number of samples that also depends polynomially on \alpha. This result is specially interesting in the case of \alpha = O(1) and the centralized setting, since the overhead is now computed based on the agnostic sample complexity. Overall, I think this is a nice paper. The problem and the setting it studies is a very natural problem. I think the extension to the non-realizable setting is also interesting. The subject matter of this paper can be of interest to the broader community and can be related to other works including, federated and distributed learning, multi-task learning, and transfer learning. One minor point is that in Algorithm R1 was discussed by Blum et al. and mentioned to have log^2(k) overhead (see last paragraph section 3 of the Blum et al.). So, the authors should mention this at some point.

Reviewer 2



This paper studies a PAC learning problem where there are k agents who each want to learn the same underlying concept. They are distinguished by the fact that they each have sample access to different distributions, and each requires the learned concept to have small expected error on their own distribution. This learning setting was introduced by Blum et al. [2017]. A bit more formally, suppose D_i is the distribution associated with agent i. The goal is to use a set of samples S (each of which is drawn from D_i for some i \in {1, ..., k}), labeled by an unknown function f*, to learn a concept f with low error. In particularly, with probability at least 1 – delta over the draw of S, across all D_i, the expected error of f should be at most epsilon. Blum et al. show that if the VC dimension of the hypothesis class is d, then there is a learning algorithm for this problem which uses O(ln^2(k) / epsilon * ((d + k) * ln(1 / delta) samples. The goal of this paper is to change the dependence on ln(k) from ln^2(k) to ln(k). The authors give two algorithms. The first algorithm has a sample complexity of O(ln(k) / epsilon * (d * ln(1 / epsilon) + k * ln(k / delta))). This algorithm weights each distribution, starting off by placing equal weight on all distributions and drawing an equal number of samples from all distributions. It finds a hypothesis with zero error on the set of samples. It then draws some more samples from each distribution to estimate this hypothesis's error on each distribution. If the error associated with distribution D_i is high, it doubles that distribution's weight. In the next round, the number of samples it draws from each distribution is proportional to the distributions' weights. It repeats this procedure for log(k) rounds. It returns the majority function over all the classifiers it learned over all the algorithm’s rounds. As the authors point out, the problem with this first algorithm is that when d = Theta(k), we again see a quadratic dependence on ln(k), as in the paper by Blum et al. [2017]. The authors get around this using another algorithm that runs for more rounds but uses fewer samples to test the error associated with each distribution. This second algorithm has a sample complexity of O(1 / epsilon * ln(k / delta) * (d * ln(1 / epsilon) + k + ln(1 / delta))). Finally, the authors also give algorithms for the non-realizable case, where there may not be a concept with zero test error, and the goal is to compete with the concept with the lowest test error. I’d like to see more discussion about how the techniques from this paper compare to the techniques of the paper by Blum et al. [2017]. Both algorithms iteratively reweight each distribution, learn a classifier over the weighted sum of the distributions, estimate the error of this classifier on each distribution, and then use this error estimate to reweight the distributions and repeat. The differences in the two papers’ approaches seem to be in the reweighting scheme, and in the number of samples needed to estimate the classifier’s error at each round. It would be helpful if the authors could summarize where the ln^2(k) factor is coming from in Blum et al.’s paper. Why does using multiplicative weights help get around this? As a smaller note, I understand the motivation of Blum et al.’s “personalized” variant of this problem, where each agent learns her own concept, but the goal is to avoid a sample complexity of O(dk). But in what scenario would the agents want to learn a single concept, which is the goal of this work? The authors give a little bit of motivation in the first paragraph, referring to AutoML, but it would be helpful to explain this connection more and provide more motivation. Finally, the notation D_i is used in Section 2 before it is defined. ===============After rebuttal================= Thanks, I like the bank loan example. I recommend adding more examples like this to motivate the paper.

Reviewer 3



This paper considers centralized collaborative PAC learning where k players collaborate to learn one classifier for all k tasks. In the realizable case, it improves the sample complexity from (ln k)^2 to ln k. It also gives an algorithm that works for non-realizable settings. The paper is clearly written, well organized, and technically sound. The results are obtained by some careful analysis and refined analysis of boosting-style algorithms. The technique used in this paper is not completely new, but I think it makes some nice contribution to the theoretical understanding of the problem. One weakness is that in the non-realizable case, the algorithm requires knowledge of OPT (the best achievable error rate) and can only obtain a multiplicative error bound. After rebuttal: Thanks for providing details on how to remove the requirement of OPT. I do think adding a remark or even putting your explanation in Appendix would make the paper look stronger.